# A paintbrush for delivery of nanoparticles and molecules to live cells with precise spatiotemporal control

Cornelia Holler [1,2,3,5], Richard William Taylor [1,2,5], Alexandra Schambony[1,2,3], Leonhard Möckl [1] & Vahid Sandoghdar [1,2,4] ✉

Delivery of very small amounts of reagents to the near-field of cells with micrometer spatial precision and millisecond time resolution is currently out of reach. Here we present μkiss as a micropipette-based scheme for brushing a layer of small molecules and nanoparticles onto the live cell membrane from a subfemtoliter confined volume of a perfusion flow. We characterize our system through both experiments and modeling, and find excellent agreement. We demonstrate several applications that benefit from a controlled brush delivery, such as a direct means to quantify local and long-range membrane mobility and organization as well as dynamical probing of intercellular force signaling.

The past two decades have witnessed impressive progress in biophysics and biotechnology, which allows more controlled and quantitative studies of fundamental cellular and subcellular phenomena. Methods such as optogenetics or laser microsurgery have made it routine to actuate mechanical or electric signals within individual cells. Advances in diverse areas such as optical microscopy, nanotechnology, microfluidics[1–3] as well as procedures such as microinjection and electroporation[4–6] make it possible to study the behavior of nano-objects or molecules within living cells.

Many biological processes are initiated by the presence of external substances such as pathogens, ligands or pharmaceutical molecules. These are oftentimes delivered via capillary blood flow and contact the cell after passing through its extracellular matrix. Conventional procedures spurt reagents into the total fluidic volume within the chamber, in an all-or-nothing way. To investigate the interaction of external reagents with the cell and their potential uptake in a quantitative manner, it is desirable to introduce them at well-defined times and locally to the close vicinity (near-field) of the cell surface. Moreover, it is highly advantageous to prevent the reagents to enter the medium surrounding the cell in an uncontrolled fashion to prevent unwanted exposure at other locations and times.

Some efforts have sought to limit the volume of the reagent by using miniaturized pipettes or atomic force microscopy tips[7–10]. However, the added substance usually remains free to diffuse in the local cellular milieu. Moreover, transfer is governed by diffusion, and so delivery is slow and stochastic. Efforts to assist in guiding reagent transport have also been pursued through application of electrophoretic forces and the use of integrated electrodes across the sample[11–13]. Another technology platform uses micro- or nanofluidic substrates, sometimes incorporating mechanical, electrical or optical elements[14]. However, this approach mandates the use of closed fluidic channels and hinders flexible access to individual cells as is common in an open dish. Furthermore, the rigid architecture of microfluidic devices makes on-command reagent delivery to select subcellular regions out of reach.

Confinement of a fluidic jet between two adjacent apertures has been used as an alternative means to confine and control reagent flux. Here, the second aperture aspires the entire flow of the first, restricting the extension of the liquid within the open aqueous environment[15–17]. This principle has been adapted for operation in microfluidic devices with large (roughly 100 μm) feature sizes[18–20], empowering applications in inactivation, staining and detachment on adherent cell cultures[20–23]. However, the large mesoscopic footprint of these devices

[1]Max Planck Institute for the Science of Light, Erlangen, Germany. [2]Max-Planck-Zentrum für Physik und Medizin, Erlangen, Germany. [3]Department of Biology, Friedrich-Alexander University Erlangen-Nürnberg, Erlangen, Germany. [4]Department of Physics, Friedrich-Alexander University Erlangen-Nürnberg, Erlangen, Germany. [5]These authors contributed equally: Cornelia Holler, Richard William Taylor. ✉e-mail: vahid.sandoghdar@mpl.mpg.de

limits instrumental access for other tools. Moreover, their large aperture constellations preclude delivery specific to individual cells within populous cultures. Micropipettes offer a pathway to address these shortcomings. An early demonstration of this approach involved an ingenious design based on concentric micropipettes[15], but it involved an intricate fabrication scheme and high fluid velocities. Similarly, the resulting lack of physiological conditions presented a limitation in alternative early efforts involving multiple largely separated micropipettes, which were used to confine the flow to the order of the size of a cell[24]. These large sizes also naturally precluded fast temporal actuation of the confinement. These drawbacks have possibly inhibited the widespread use of this approach in cell biology over two decades.

To overcome previous shortcomings, we have devised a simple-to-implement and inexpensive design to achieve physiological contact-free reagent delivery onto and through the surfaces of individual cells with micrometer spatial accuracy and millisecond temporal resolution. Figure 1a illustrates the design concept and device features for confining a femtoliter volume at a positionable tip. By gently bringing a target surface into the confined envelope of laminar flow with velocities in the order of mm s$^{-1}$, we administer minute amounts of reagents into the cellular near-field, with a particular attention to the integrity of the nanoscopic entities such as the glycocalyx or cilia at the exterior of the plasma membrane.

In what follows, we characterize the performance of the methodology, which we refer to as micro-kiss (μkiss), and demonstrate how its spatiotemporal command provides numerous pathways for new and more exact investigations into cellular processes. Our procedure is fully compatible with conventional open-dish preparations, requires no microfabrication steps (Supplementary Section 1) and can be easily adopted in most biologically oriented laboratories.

## Results and discussion

The μkiss brush comprises a pair of microfluidic apertures arranged in near-touching contact. The scanning electron microscopy (SEM) images in Fig. 1b show a realization using two heat-pulled glass micropipettes. In this example, the apertures have an inner diameter of $\varnothing_{in}$ = 6 μm and a wall thickness of 1 μm. The minimal footprint of the micropipette pair permits implementation of this in virtually any commercial or custom-built setup. By bending the micropipettes with a high aspect ratio taper, one can also operate comfortably in the 2.5 mm working space between the sample and a water-dipping objective of an upright (confocal) microscope, as was done in the current study. The exact geometry of the micropipette system (aperture opening, bend angle, taper and so on) can be adjusted at will for each specific instrumentation and measurement (Supplementary Section 1).

Figure 1c illustrates the simulated steady-state flow field lines for the arrangement shown in Fig. 1b. The injected reagent is confined to a tight envelope of flow between the two apertures (blue lines in Fig. 1c) as pressure-driven laminar flow is injected at rate $Q_{inj}$ out of a storage chamber and is subsumed within the larger counter flow of the aspiration aperture at rate $Q_{asp}$. To ensure flow confinement, it is required that $Q_{inj} < Q_{asp}$. Empirically, we find $Q_{ratio} = \frac{Q_{inj}}{Q_{asp}} < 0.8$ leads to a reliable confinement. For an aperture diameter of 1 μm, the volume of the perfused envelope amounts to approximately 0.35 fl. The compact envelope and small in-plane angle between the micropipettes allow us to place the apertures in vicinity of the desired location. We emphasize that the amount of material that is painted onto a surface is usually much less than the volume of the envelope.

Confinement of reagents to a fluid jet results when advection in-stream dominates over diffusion, a situation characterized by a large Péclet number (Pe ≫ 1). To verify this within our system, we examined the advective flow velocities present in the brush envelope. In Fig. 1d, we present the outcome of numerical simulation of mass transport for the arrangement shown in Fig. 1b. In Fig. 1e, we show the experimental

results of high-speed tracking (10,000 frames per second) of single colloidal gold particles (80 nm in diameter) as probes using interferometric scattering (iSCAT) microscopy[25] (see Supplementary Section 2 for technical details). By localizing the in-plane position of a probe particle every 100 μs, the step-to-step differential velocity of the flow can be calculated. The resulting flow trajectories of 200 particles indicate excellent agreement between the model and experiment.

We verify that within the spherical envelope, flow speeds decrease with increasing distance from the apertures, with the region near the aspiration aperture consisting of larger velocities owing to the higher inward flow rates. We also note that velocities typical in the outward-most regions are in the range $v$ = 1–2 mm s$^{-1}$, comparable to physiological blood flow rates in the body, especially in capillaries. In Fig. 1f, we plot the calculated velocity distributions along the axis through the center of the envelope for various flow rate ratios, $Q_{ratio}$. As $Q_{ratio}$ is decreased, that is, when the aspiration is increasingly stronger than injection, the velocity distribution is shifted to larger values.

Figure 1g shows that the extent of the μkiss brush stroke, which in turn scales with the perfusion envelope size, can be tuned through adjusting $Q_{ratio}$. In other words, it is $Q_{ratio}$ that is critical for determining the envelope size and not the absolute constituent flow rates. Given the direct relation between the envelope size and the aperture diameter, it is thus desirable to minimize the latter. We find micropipettes with internal diameter $\varnothing_{in}$ = 1 μm to be the narrowest dimensions that can be conveniently handled without substantial hydraulic resistance to flow. We believe this brush size is sufficiently small for most applications in cell biology.

We calculate for matter with diffusivities in the range $D$ = 10$^{-9}$–10$^{-12}$ μm$^2$ s$^{-1}$ (corresponding to a size range of 1–500 nm), and Péclet numbers in the range Pe = 10$^1$–10$^3$ for the flow parameters considered in this work ($Q_{inj}$ = 0.03–3 nl s$^{-1}$), confirming the dominance of advection as the principle means of mass transport. Our calculations implicitly assume the reagent fluid to be water (with viscosity $\mu$ = 0.9544 mPa s, for temperature $T$ = 22 °C), but other solvents may also be used. Further information regarding numerical simulations and characterization of the brush performance can be found in Supplementary Sections 1 and 3).

As with a paintbrush, one can form strokes during deposition for patterning applications. In Fig. 1h, we demonstrate brushing 200 nm fluorescent beads onto a glass coverslip, painting the acronym 'MPZPM' for the Max-Planck-Zentrum für Physik und Medizin as a complex pattern. We verify that the average feature size of the written pattern is less than or equal to 10 μm, comparable to the typical working envelope from a micropipette with $\varnothing_{in}$ = 6 μm.

Our numerical modeling reveals shear stresses in the flow to be in the range of −0.3 < $p$ < 0.1 Pa at the boundary (Fig. 1i). This is substantially smaller than those exerted in similar scenarios, such as in the micropipette aspiration technique[26], which can exceed $p$ = 10$^2$–10$^3$ Pa, and which are applied for considerably longer periods. One also observes in Fig. 1i a region in the mid-point of the envelope where the two opposing pressure gradients cancel, coinciding with the section used for sample interaction and targeted delivery. The minimal shear stress associated with the envelope reaffirms the viability of the technique for live cell investigations, which we further verified through a LIVE/DEAD assay on a sample of ten cells (Supplementary Section 4).

Reagent matter is transferred out of the confinement envelope when it overlaps with a target surface to which the reagents can bind[27]. The dimensionless Damköhler number, $D_N$, characterizes whether the process of surface adsorption is limited by the mass transport of the reagent to the surface ($D_N \gg 1$) or by the affinity between reagent and surface ($D_N \ll 1$)[28,29]. An effective μkiss brush is one in which the latter dominates. To verify that our high-Péclet flows result in a large mass flux onto the surface[30], we estimate $D_N$ for a model system of labeling receptors on the cell membrane, following the framework of ref. 29. With knowledge of Pe at the envelope boundary, and assuming a surface receptor site density of roughly 100 per μm$^2$ and an association rate constant $k_{on}$ = 3 × 10$^3$ M$^{-1}$ s$^{-1}$ (that of Cholera toxin subunit B (CTxB)

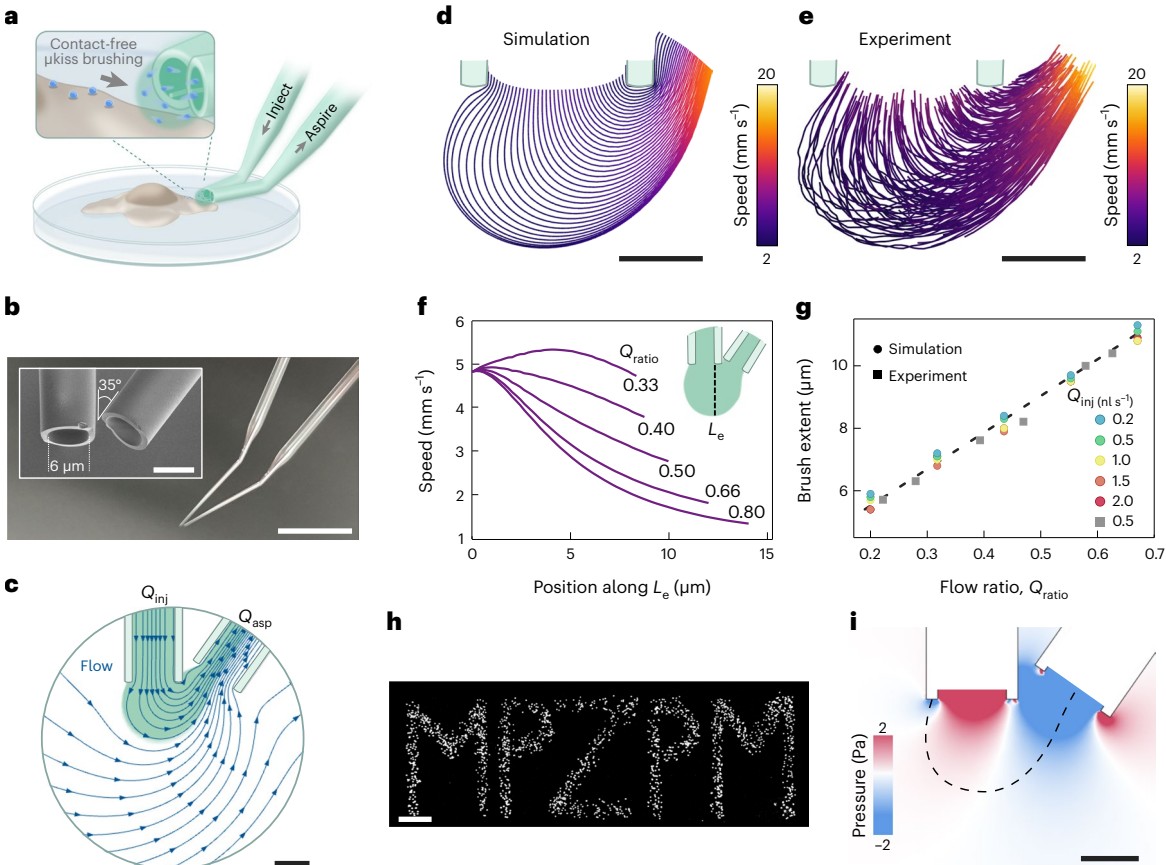

**Fig. 1 | Flow confinement within the μkiss paintbrush. a**, Schematic of a micropipette pair for coating live cells. **b**, Image of the micropipette pair with an inset showing a SEM image of the tapered micropipettes with an inner diameter of 6 μm and their angular displacement of 35°. Scale bar 5 mm, inset 5 μm. **c**, Numerical simulation of the confined flow (green) with the direction of fluid streamlines overlaid (blue) for the geometry in **b**. Scale bar 5 μm. **d**,**e**, In-stream flow lines and velocities for gold nanoparticles (diameter, 80 nm) in simulation (**d**) and experiment (**e**) for $Q_{inj}$ = 0.11 nl s$^{-1}$ and $Q_{ratio}$ = 0.27. Scale bars

5 μm. **f**, Simulated stream velocities along line $L_e$ shown in the inset for various flow rate ratios $Q_{ratio}$ at $Q_{in}$ = 0.16 nl s$^{-1}$. **g**, Envelope extent for various flow ratios and absolute volumetric flow in both simulation and experiment. **h**, Image of a pattern generated on a glass coverslip by controlled deposition of 200 nm fluorescent beads. Scale bar 20 μm. **i**, Simulated shear stress in the vicinity of the apertures. Dashed lines mark the envelope boundary for $Q_{inj}$ = 0.16 nl s$^{-1}$ and $Q_{ratio}$ = 0.3. Scale bar 5 μm.

binding GM1 (i.e. CTxB-GM1) (ref. [31]) used later), we find $D_N = 10^{-3}-10^{-6}$. This confirms that we operate in the reaction-limited regime. Hence, surface adsorption performance is independent of our flow rate[30].

## Diffusion on a model membrane

Quantification of membrane dynamics and transport properties is usually accomplished through binding an optical label to the molecule of interest. Fluorescence recovery after photobleaching (FRAP) is the most widely used method to measure membrane diffusion. In this approach, a fluorescently labeled membrane is locally bleached in a fast process through application of a strong laser beam. By analyzing the signal replenishment in video recordings, one can extract the diffusion constant of the mobile species. Although effective and convenient, FRAP has some disadvantages that should be kept in mind. First, a very high level of global labeling is required. Second, application of intense laser light can cause photodamage and toxicity. Third, it is nontrivial to identify local events, for example, at nano-domains. To remedy these deficits, trajectories of single molecules and nanoparticles have been recorded and analyzed[32,33]. However, even in this case, delivery of these nanoscopic entities to the membrane has been achieved in a uniform fashion, that is, by adding a macroscopic quantity of material to the buffer and incubating over timescales of minutes or longer. We now show that μkissing can be used to address the shortcomings confronted by FRAP and conventional single-particle tracking.

For the purpose of a general demonstration, we picked an established synthetic model membrane system comprising the GM1 lipid embedded within a fluid 1,2-dioleoyl-*sn*-glycero-3-p hosphocholine (DOPC) and 1,2-dioleoyl-*sn*-glycero-3-phospho-L-serine (DOPS) lipid bilayer as illustrated in Fig. 2a. To fluorescently label GM1, we used an AlexaFluor-conjugated CTxB (CTxB-AF488), which binds the oligosaccharide moiety of up to five GM1 with high affinity[34,35].

The sequence of our μkiss brushing is shown in Fig. 2b(i)–(iii), with corresponding fluorescence microscopy of labeling shown in Fig. 2c(i)–(iii). First, the micropipette pair is lowered until the full width of the aperture, which contains fluorescent material, is in focus. For this measurement, we use a micropipette pair with apertures $\varnothing_{in}$ = 1 μm to achieve a labeling area comparable to a diffraction-limited focus. The depth of focus of our ×40 water-dipping microscope objective is 0.65 μm, allowing us to avoid contact with the surface. An expanded account is provided in Supplementary Section 1. Pressure-actuated flow is then initiated between the two apertures, beginning with the aspiration channel, creating the CTxB-AF488 envelope of 100 μg ml$^{-1}$ on the membrane, sustained for one imaging frame (130 ms). On the next imaging frame, flow is ceased to halt further labeling. The sample is then imaged via confocal scanning microscopy with the objective positioned above the membrane. The temporal instrument response function, that is, the

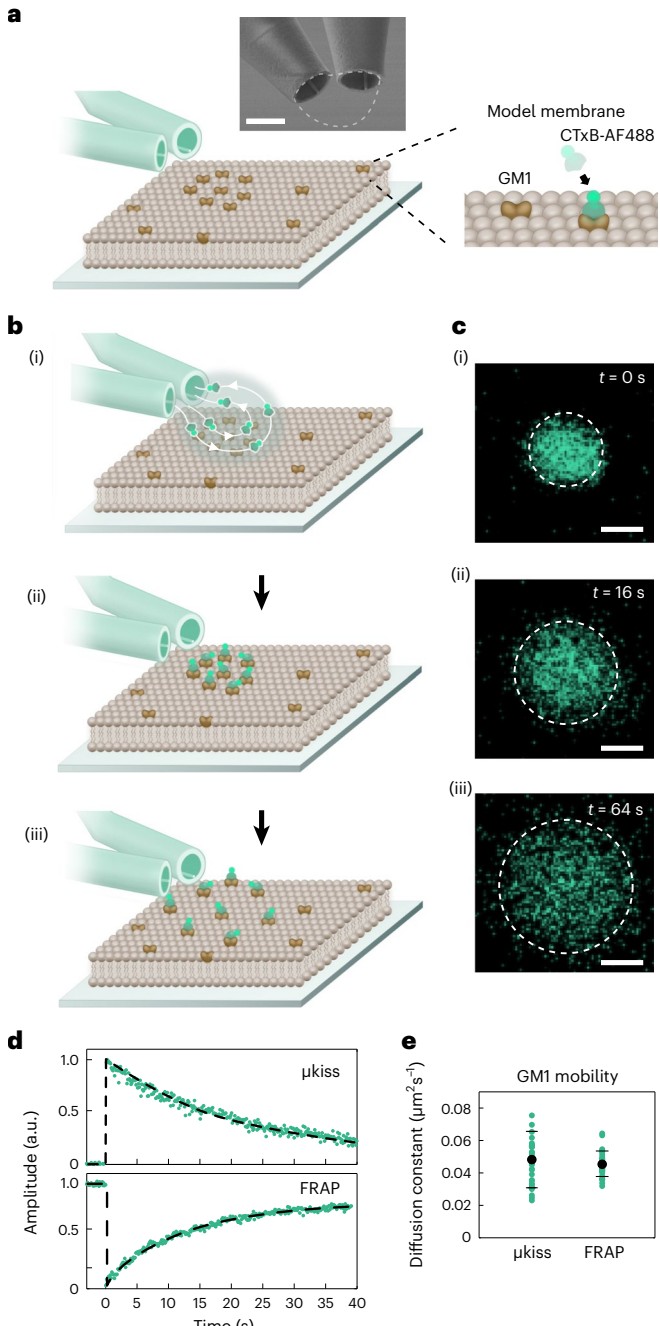

As displayed in the upper panel of Fig. 2d, the diffusion of the labeled GM1 leads to a decrease in the fluorescence signal at the delivery location. The signal can be directly analyzed to extract a mean diffusion coefficient of $D_{\mu kiss} = 0.048 \pm 0.017\ \mu m^2\,s^{-1}$, plotted in Fig. 2e. In Fig. 2d,e, we also present the outcome of in situ FRAP measurements performed on the same sample (Supplementary Section 6) following the analysis presented in ref. 36. Here, the diffusivity was calculated through extracting the rate of change in label signal intensity with (careful) knowledge of its initial distribution. We obtain a mean diffusion constant of $D_{FRAP} = 0.045 \pm 0.008\ \mu m^2\,s^{-1}$, which is both in agreement with reported values from similar measurements on this membrane system[37] and with our μkiss result from a series of $n$ independent measurements, here with $n = 30$ (Fig. 2e). However, the precision of the latter is lower as evidenced by its slightly larger error bar. This results from a fivefold lower signal due to our wish to image with a shorter exposure time. As shown below, adjustment of the concentration of the desired marker in the injected liquid allows one to tune the labeling density down to the single-particle level, thus, uniting ensemble (for example, FRAP) and single-molecule studies in one approach.

### Subcellular targeting of individual cells

**Receptor lipid diffusion in the plasma membrane.** As demonstrated with synthetic bilayer membranes, μkiss labeling can be used to investigate mobility within the plasma membrane of the live cell, which we demonstrate in Fig. 3a–d for the GM1 lipid in COS-7 cells. Again, we use a micropipette pair with an aperture $\varnothing_{in} = 1\ \mu m$ to perform 'punctual labeling' during one imaging frame (duration 0.7 s, Fig. 3b(i)). As displayed in Fig. 3b(ii)–(iv), the labeled lipids quickly spread through diffusion. A video is provided in Supplementary Video 1. Analysis of the fluorescence signal from a series measurement ($n = 8$) reveals a diffusion constant of $D_{\mu kiss} = 0.32 \pm 0.13\ \mu m^2\,s^{-1}$ for GM1-CTxB-AF488 (see Supplementary Section 6 for details). FRAP measurements performed on the same sample report $D_{FRAP} = 0.35 \pm 0.14\ \mu m^2\,s^{-1}$, which is in line with previously published values[38]. The detailed structures in Fig. 3b illustrate a decisive advantage of μkiss, which not only yields a value for the diffusion constant but also reveals the presence of domains in distinct membrane regions[39].

The μkiss brush can also be used to deliver the reagent in a continuous mode to monitor long-range diffusive phenomena within the membrane. An example of such an 'extended labeling' is illustrated in Fig. 3c. Here, we used a micropipette pair with larger openings ($\varnothing_{in} = 6\ \mu m$), but we positioned them at a distance 2 μm away at the very periphery of the cell so that only GM1 molecules within a thin crescent within reach of the μkiss brush are labeled. In this manner, we observed diffusion of GM1 lipids over length scales longer than what has previously been possible for plasma membranes. In Fig. 3d, we plot the time evolution of the fluorescence intensity signal along the dashed line marked in Fig. 3c(iii). The bright line-like features in Fig. 3d correspond to dye-enriched clusters. The mobility of one such cluster (marked with dashed line) is found to have a linear speed of 7.5 nm s$^{-1}$. A video of the labeling in Fig. 3d is provided in the Supplementary Video 2a along with intensity line cuts in Supplementary Video 2b.

**Response of a cell to a local stimulus.** Another interesting application of our methodology is in the investigation of how the cell responds to toxic or otherwise antagonistic substances. For a simple illustration, we picked demecolcine, a membrane-permeable antineoplastic drug that disrupts the microtubule network within the cell by depolymerizing existing tubules and limiting their subsequent reformation[40]. To visualize the microtubule network in COS-7 cells, we transfected a cell culture to express an α-tubulin-pmEGFP fusion protein. Figure 3e displays the fluorescence image of the microtubules in a cell, and Fig. 3f–h shows a close-up of the region of the dashed box in the Fig. 3e, which we monitored over about 9 min after a μkiss. As seen in Fig. 3g, 3 min after continuous application of 27 μM demecolcine, the microtubules were shortened in the vicinity. Following cessation of delivery

**Fig. 2 | μkiss labeling and lipid diffusion in a model membrane. a**, Schematic showing a synthetic membrane of DOPC–DOPS with a low concentration of GM1 approached by the brush micropipette pair. The close-up shows an impression of clustered GM1 and its labeling by CTxB-AF488. Inset shows SEM of the $\varnothing_{in} = 1\ \mu m$ micropipette pair, with the μkiss envelope shown by the dashed line. Scale bar, 1 μm. **b**, Steps of a diffusion experiment. First, the flow is switched on, depositing CTxB-AF488 on the membrane (i). After 1 s of operation, flow is switched off (ii) and CTxB-AF488-labeled lipids diffuse (iii). **c**, Confocal fluorescence image series of the sequence in (**b**(i–iii)), representative of a recorded series of size $n = 30$. Scale bars 1 μm. **d**, Peak fluorescence amplitude of a μkiss labeling and FRAP measurement as a function of time. a.u., arbitrary units. **e**, Measured diffusion constants for μkiss and FRAP (green) with their mean and standard deviation (black), each from a series of size $n = 30$.

time it takes for the flow to be turned on or off, depends on the applied pressure and micropipette opening and can be as fast as 35 ms (Supplementary Section 5).

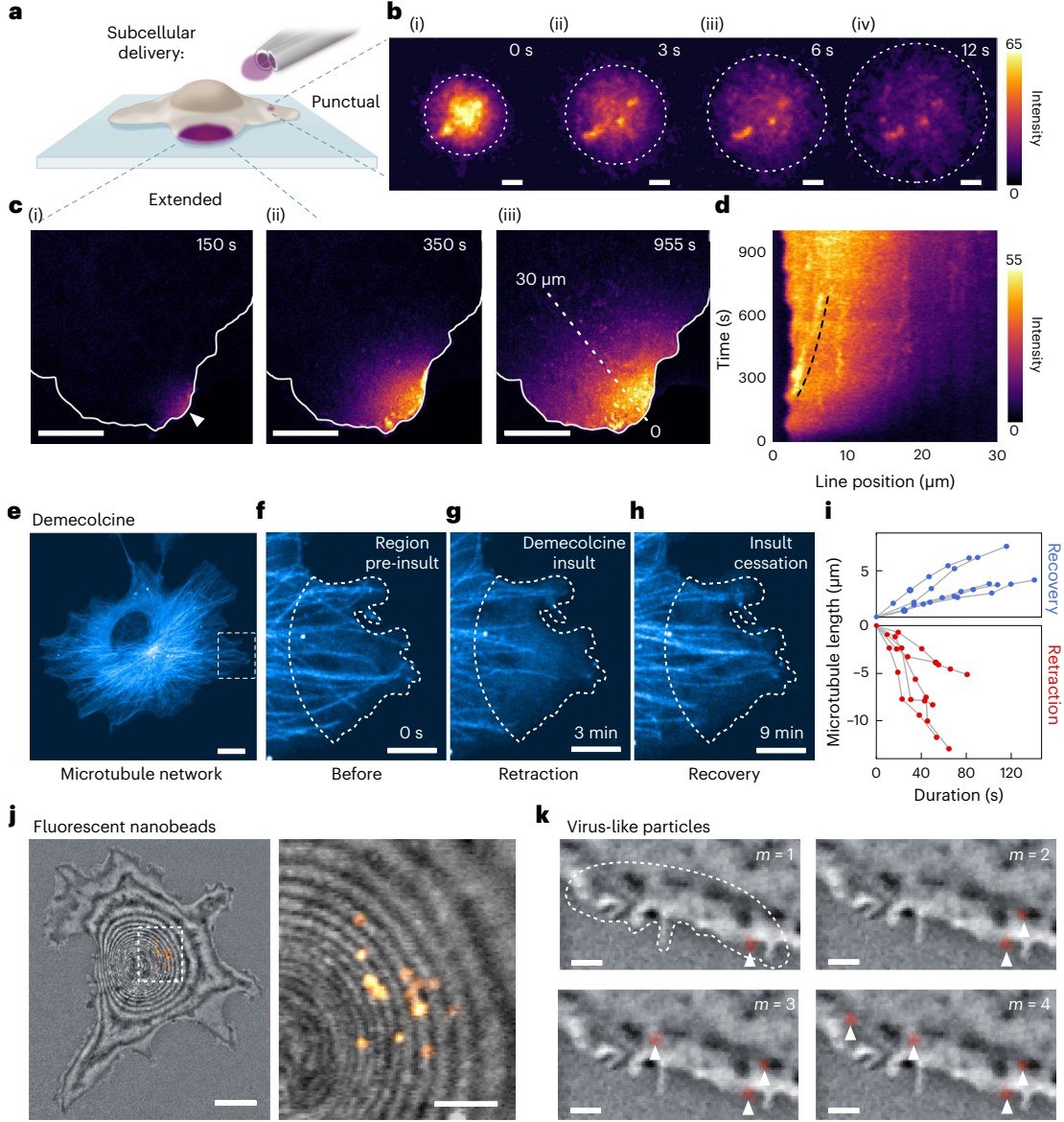

**Fig. 3 | Subcellular delivery to live cells. a**, Illustration of punctual and extended labeling of a cell. **b**, Fluorescence images of a region of a COS-7 cell membrane, where GM1 lipids are punctually labeled by CTxB-AF488 at the time of labeling (i) and 3, 6 and 12 s subsequently ((ii)–(iv), respectively). Measurement representative of a recorded series of size $n = 8$. Scale bar 1 μm. **c**, Fluorescence images of the 'extended' membrane labeling of GM1-CTxB-AF488, following 150 (i), 350 (ii) and 955 s (iii) of continuous labeling. Scale bars 10 μm. **d**, Fluorescence signal along the dashed line from **c** as a function of time. Dashed lines guide the eye. **e**–**h**, Demecolcine treatment of microtubule network. **e**, Fluorescence image of pmEGFP-labeled microtubule network. Scale bar 10 μm. **f**, Close-up view of demecolcine target region (dashed line) before exposure. **g**, Retraction of microtubules in the target area following continuous demecolcine exposure. **h**, Same region as **f**, 6 min after cessation of demecolcine delivery. Scale bars **f**–**h**, 2 μm. **i**, Kymograph visualizing the retraction and recovery of several chosen microtubules in the target area following the cessation of demecolcine application over time. Details of analysis provided in Supplementary Section 7. Measurement representative of recorded series ($n = 5$). **j**, Composite image showing a cell via confocal iSCAT (grayscale) and delivery of fluorescent beads (orange, 200 nm) to the membrane at the apical center of the cell (left), with close-up view (right). Scale bars 10 μm (left) and 5 μm (right). **k**, Particle-wise delivery of $m$ individual fluorescent virus-like particles (red) in time on a cell (confocal iSCAT). Scale bars 1 μm.

at 4 min, microtubule reformation is observed (Fig. 3h). Figure 3i presents a plot of the demecolcine-induced retraction and recovery of five microtubules within the region of interest, whereby we plot the abstracted changes in microtubule length. A video of this process is provided as Supplementary Video 3. We verified that under similar application of the μkiss with a nondisruptive agent, the retraction-recovery behavior of the microtubule network was not observed (provided in Supplementary Section 7).

We now transition from small molecular reagents to those containing nano(bio)particles. Figure 3j shows the localized delivery of 200 nm

carboxylate-coated fluorescent beads, which readily adhere to the cell membrane. The simultaneously recorded fluorescence and confocal iSCAT[41,42] images of a COS-7 cell are overlaid. The close-up of the region marked in the left panel of Fig. 3j is presented in the right panel of this figure and shows that about a dozen individual nanoparticles were deposited within a small region at the apical dome of the cell.

An important application of nanoparticle delivery is in quantitative virology research[43], where it is desirable to study the mechanisms of entry, assembly and egress with single-virus precision[44]. In Fig. 3k, we demonstrate stepwise delivery of individual Atto488-labeled

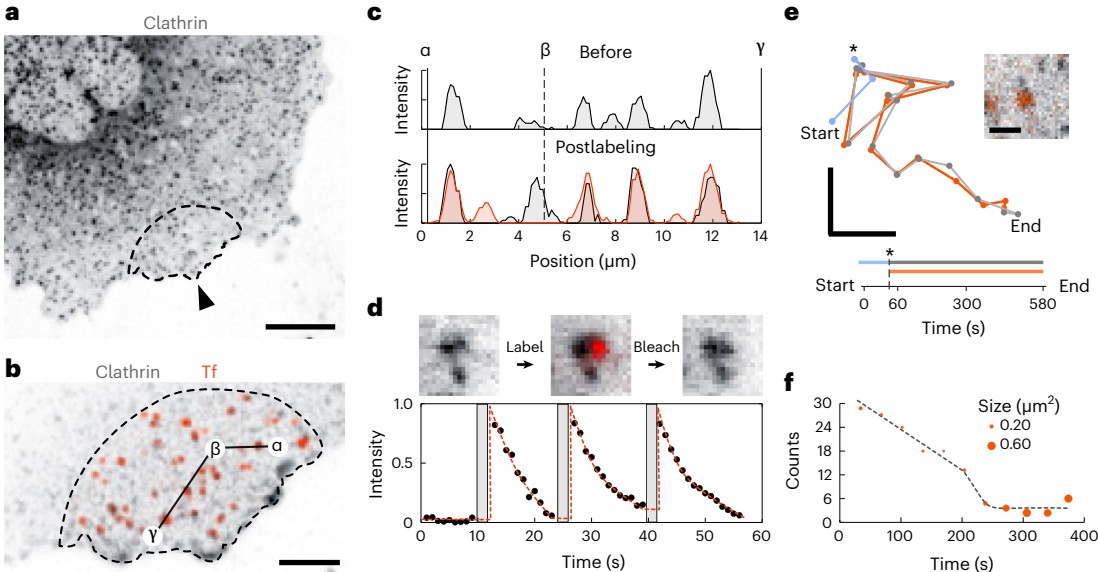

**Fig. 4 | μkiss labeling to visualize time-dependant TfR-clathrin membrane dynamics. a**, Fluorescence image of HaloTag-OregonGreen-labeled clathrin. The arrowhead and dashed outline mark the labeling region. Scale bar 10 μm. **b**, Close-up view of the region in **a** postlabeling, showing clathrin (black) and Tf (orange). The result is indicative of behavior from a series of size $n = 13$. Scale bar 5 μm. **c**, Profile of fluorescence signal along the path αβγ marked in **b**, before Tf labeling (above) and after (below). **d**, Images of one Tf-labeling cycle on an immobile clathrin cluster (upper) and integrated signal of Tf fluorescence for three labeling cycles (bottom). **e**, Trajectory and time-schematic of a mobile clathrin spot during flight before (blue) and after (gray) Tf labeling. The trajectory for the Tf-labeled TfR is highlighted in orange. An asterisk marks the point of labeling. Scale bars 1 μm. The inset shows the tracked clathrin cluster with colocalized Tf signal. Scale bars 1 μm. Lower axis marks the temporal ascension of time. **f**, Number and size distribution of Tf-labeled TfR clusters as a function of time, following initial TfR labeling.

adenovirus-like particles from the Johnson & Johnson COVID-19 vaccine over the duration of 550 s by using a concentration of $3 \times 10^8$ VLPs per ml. Although the flux of particles from the perfusion volume to the surface is a stochastic process across the targeting zone, the particle concentration can be sufficiently lowered to enable on average single-particle delivery. If greater definition over the labeling zone is required, one can also use the point-labeling strategy with narrower micropipettes.

## Time-dependent membrane organization

As mentioned earlier, a decisive advantage of the μkiss brush technology is access to time-sensitive processes that are triggered through binding. One such example is membrane organization and signaling, wherein the binding of a ligand to a membrane receptor initiates a complex sequence of physico-chemical interactions, guided by the organizing principles of the membrane. A well-established example of this principle is the transferrin receptor (TfR), which facilitates cellular uptake of iron. The binding of iron-rich transferrin (Tf) to the receptor results in fast internalization of the ligand–receptor complex, with a great wealth of data indicating that internalization proceeds through clathrin-mediated endocytosis[45]. Nevertheless, published reports point toward the importance of the binding site, as accounts differ regarding how TfR and clathrin are mutually organized before the point of ligand binding[46]. We demonstrate here how μkiss labeling can be applied to help address this question.

To visualize clathrin within a live COS-7 cell, we used the HaloTag strategy to fluorescently label overexpressed clathrin light chain with an OregonGreen fluorophore, shown in Fig. 4a. We performed labeling with a 1 mM Tf-AF647 solution on the dashed region highlighted in Fig. 4a for a single period of 30 s. We imaged the process in both color channels before, during and after Tf labeling. The latter is shown in Fig. 4b. An estimate of Mander's colocalization coefficients within the labeling region yielded values of 0.71 and 0.62 for Tf-in-clathrin and clathrin-in-Tf cases, respectively. More details on Tf labeling are

provided in Supplementary Section 8, and a video is presented in the Supplementary Video 4.

To inspect the distribution more closely, in Fig. 4c we plot the signal intensity along the path αβγ in the frames preceding and following labeling. We observe that Tf predominantly labels regions on the membrane with a pre-existing clathrin presence, but not exclusively. The strength of our approach is in the spatial-temporal control of labeling, which allows us to gain insights into receptor densities within the membrane. Through calibration, we quantified the TfR clusters to contain a distribution between 2 and 20 TfR per cluster (Supplementary Section 9), a result consistent with similar reports on human embryonic kidney cells[47]. We emphasize that bulk quantification of this quantity is, to this day, not a trivial task, let alone examination at the single-cell level such as demonstrated here.

Another advantage of our localized labeling strategy is that outside the target region the membrane remains in its native, prelabeled, state. This enables further labeling experiments on the same cell. In addition, within the target region, one may also label repeatedly the same structures. We illustrate so in Fig. 4d, wherein we identified an immobile clathrin cluster and performed multiple cycles of Tf labeling and bleaching (upper panel). Swift Tf labeling was achieved within one imaging frame (1 s) and we illuminated until the labeled signal was bleached. The integrated fluorescence signal shown in the upper panel of Fig. 4d is plotted for three cycles in the lower panel, demonstrating the repeated labeling of the same membrane patch. We note that such back-to-back labeling is possible, as we are not saturating the available TfR molecules within each short labeling window.

Not all clathrin spots in the plasma membrane are immobilized, with a small fraction exhibiting lateral motion[48]. In Fig. 4e, we track one such clathrin spot before (blue) and after (gray) labeling colocalized TfR with the Tf ligand. We mark the labeling time, $t = 0$ s, mid-flight, with an asterisk. We find that after labeling, the Tf–clathrin complex exhibits correlated diffusion and increasingly directed motion over time consistent with the rate of Tf uptake[45]. These data indicate activation of

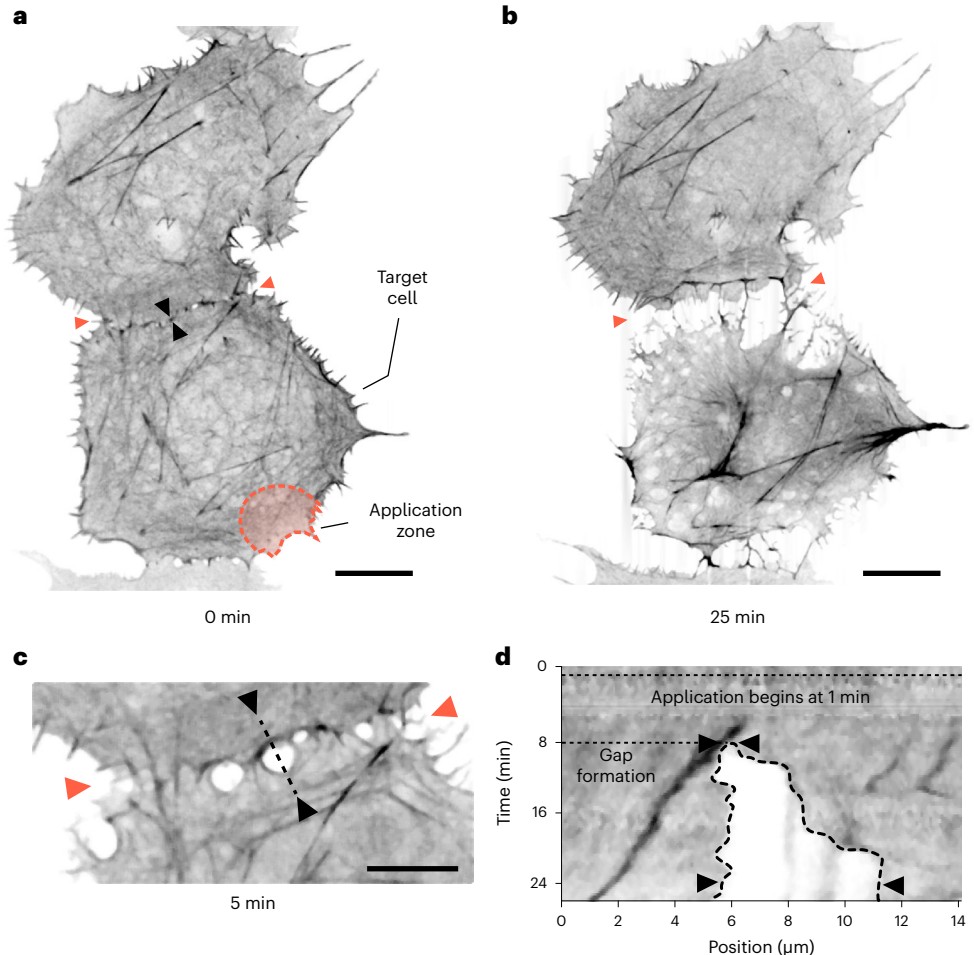

**Fig. 5 | Visualizing intercellular forces at a junction. a**, Fluorescence image of a linear ensemble of cells with mEmerald-labeled actin filaments. Targeted cell and application zone are marked. **b**, Image of two cells following Lat-A treatment for 25 min to the application zone shown in **a**. Orange arrowheads act as reference markers throughout the figure for the cell–cell boundary. Black arrowheads denote a region of gap formation, discussed later. Observation is indicative of results obtained from a series of two measurements. Scale bars 10 μm. **c**, Close-up view of the cell–cell contact boundary 4 min after first exposure to Lat-A. Black arrows mark the site of gap formation. Scale bar 5 μm. **d**, Kymograph showing an increase in the gap size over time along the line marked in **c**.

TfR, leading to local clustering within clathrin-coated pits over time, which eventually become internalized.

In a final study, we labeled a pristine membrane patch and monitored the evolution of the cluster density within a $15 \times 15\ \mu m^2$ region. Figure 4f presents the number and size of clusters as a function of time. We find that after several minutes, smaller clusters begin to coalesce into larger mobile ones, reducing their overall number. After 5 min, the large clusters dominate. These enlarged Tf clusters remain laterally mobile and frequently colocalize with clathrin. This observation is consistent with established reports for the fast internalization (roughly $k_{int} = 0.20\ min^{-1}$) and recycling process of the Tf–TfR complex[49]. The spatiotemporal control of μkiss brushing is key to access such early dynamics. In summary, the results indicate that a large fraction of membrane-localized TfRs either reside in a pre-assembled state in clathrin-coated pits or are rapidly shuttled toward them upon binding to Tf.

## Probing multicellular interactions

As quantitative nanobiophysical studies advance, subcellular and single-cell methods are also being used to investigate a plethora of cell–cell interactions[50], especially those that involve mechanical signaling[51]. A prominent body of research has been devoted to how cytoskeleton reorganization and disruption affect individual cells, as well as their interactions with their environment[52]. Equally relevant, but experimentally much more challenging to implement, is the examination of network effects within multicellular systems such as the mechanism for the propagation of cytoskeleton disruption between neighboring cells. Clever realizations have recently been implemented using optogenetics[53]. However, (bio)chemical stimuli would be more desirable since they emulate more closely life processes. In addition, approaches such as those presented in our work require no previous treatment of the cells and are, thus, more generally applicable, especially in medical studies of primary cells.

In Fig. 5, we illustrate the application of the μkiss brush for introducing chemical structural perturbations to a particular cell coupled with another. Here, we compromised the actin cytoskeleton of the target cell by local administration of latrunculin-A (Lat-A). Lat-A is a membrane-permeable toxin and binds actin monomers near their nucleotide binding site, preventing their polymerization and disrupting existing filaments[54]. To visualize the actin network in COS-7 cells, we transfected a culture to express the Lifeact peptide labeled with mEmerald that binds and fluorescently labels filamentous actin.

Figure 5a presents an image of fluorescently labeled actin filaments from a cellular ensemble wherein the cells are in their normal state before targeting (at time, $t = 0$ min). In Fig. 5b, we show the result of 20 μM Lat-A applied to the marked target cell for 25 min. As expected,

disruption of the force-carrying actin network leads the cell to inwardly retract. We find that this occurs in directions normal to a boundary communicating with another cell, an observation that is consistent with the distribution of forces at the cell–cell boundary[55]. A complete time-lapse video of the actin degradation sequence is provided in Supplementary Video 5, and the resulting cell area change is given in Supplementary Section 10. Figure 5c,d highlights a rupture forming at the cell–cell boundary 5 min after Lat-A exposure began. The dynamic separation of the gap is presented as a kymograph for the intensity along the marked line in this figure, revealing a staggered and feature-rich response as the targeted cell begins to yield. Studies such as that shown in Fig. 5 could also be performed using conventional microinjection strategies. However, the intrusive nature of puncturing the cell creates difficulties in attributing the response of the cell to that which was intended.

## Conclusion and outlook

We have presented the μkiss brush for contact-free delivery of small molecules (low-mass fluorescent dyes such as Alexa Fluor488) and (bio) nanoparticles (beads, virus-like particles) to select micrometer-sized regions in the near-field of live cells. With this approach, one can mimic the initial encounter and interaction of the reagent with a cell in a near-to-physiological fashion and yet achieve a remarkable degree of spatial and temporal control. We demonstrated the performance of the approach by local labeling of synthetic lipid bilayers as well as plasma membranes of live cells in the context of a wide range of investigations of current interest. We showed that in addition to a spatial control of about 1 μm, the brush can be activated at a high temporal resolution in the order of 35 ms, allowing one to deliver the reagent of choice 'on the fly', that is, during a particular event under observation. Notably, our technique also lends itself to measurements carried out on cells within a complex setting such as tissues, organoids or small organisms, for example, to achieve cell-cycle-specific analysis in a nonsynchronized cell culture. We emphasize that our contact-free delivery demonstrated here cannot be achieved with a sole nozzle-like structure, where the spurted jet of reagents is unconfined and leads to a large labeling area, even when submicrometer-sized apertures are used (Supplementary Section 11).

The knowledge and control of the concentration of the species and their flow rate covered here spans several orders of magnitude, including the regime of physiological flow rates encountered in the capillaries of the body. Such implementations predict use in spatially resolved quantitative investigations of binding affinity[56], endocytosis[57], localized mechano-transduction and transmission of forces across skeletal networks[58]. With careful design of the flow envelope and associated shear forces, probing of the binding interaction between the membrane and entities such as viruses[59], extracellular vesicles[60] and nanoparticles[25] is also foreseeable. The realization of the μkiss brush is inexpensive, and the minimal footprint of the micropipette pair permits facile incorporation within virtually any microscope common to the biologically oriented laboratory. In the future, smaller envelopes can be engineered by reducing the size of the apertures, although this might require new ideas since the rapid increase in hydrodynamic resistance makes it challenging to achieve substantial flows. Moreover, the method would benefit from solutions for minimizing the adhesion of the sample reagents onto the micropipette walls to lower the rate of clogging as well as loss of sample reagents. Another route for improvement could exploit strategies for fabricating monolithic multi-aperture structures and with more nanoscopic control of their morphology.

## Online content

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

## Methods

### Numerical simulations

Finite element simulations were performed with COMSOL Multiphysics v.6.0, combining modules of single phase flow, transport of dilute species in stationary and transient two- and three-dimensional models. Further information is provided in the Supplementary Section 1.

### Microscopy

Confocal imaging is performed with a Zeiss LSM 800 fitted with a water-dipping objective (×40, numerical aperture (NA) 1.0). The scanning confocal microscope allows for large field-of-view imaging of the cell culture, and the working distance of 2.5 mm permits ready access to the cell culture in the focus. The limited amount of space between objective and sample normally precludes the use of additional probes or instruments, but here we demonstrate our system can fully operate in this available gap.

The sample may be excited with illumination wavelengths $\lambda = 405$, 488, 561 and 640 nm. Through appropriate setting of the wavelength-selective filter, the sample can be imaged in a bright-field and fluorescent configuration.

Widefield iSCAT imaging of the sample is performed with the lower oil-immersion objective (×100, NA 1.4). Laser light from a supercontinuum white light laser (NKT Photonics) in the range of 450–900 nm is filtered down to $\lambda = 550 \pm 15$ nm through a Varia filter box and focused onto the back focal-plane of the objective to give a field of view of $10 \times 10$ $\mu m^2$. The sample is imaged onto a high-speed camera (Vision Research, Phantom, Miro LAB v.3a10) with a resultant 277-fold magnification.

The experimental arrangement for simultaneous confocal and widefield imaging is accomplished through two objectives positioned above and beneath the sample, respectively.

SEM was performed for microstructural characterization of the glass micropipettes. SEM images have been recorded with a Hitachi S-4800 High Resolution SEM using an acceleration voltage of 2 kV.

### Plasmids

Plasmids were purchased from Addgene: mEmerald-Lifeact-7 (54148) and pmEGFP-α-tubulin-IRES-puro2b (21042). pSems-ClathrinLC-Halotag plasmid was a gift from J. Piehler (University of Osnabrück). Transfection was performed with Lipofectamine 3000 Transfection Reagent (Invitrogen) according to the manufacturer's protocol.

### Micropipettes

Micropipettes were purchased from Eppendorf. We used a Piezo Drill Tip M.ICSI with inner aperture diameters of 6 μm.

Micropipettes with inner aperture diameters of 1–2 μm were fabricated in-house using borosilicate glass capillaries with an in-built filament (Science Products GmbH, catalog no. GB100TF-10). Micropipettes were pulled with a laser-based P-2000 Micropipette Puller from Sutter Instruments. Further details on the pulling protocol are given in Supplementary Section 1.

Before use, the injection micropipette was filled with the diluted material via a microcapillary loader pipette for ICSI micropipettes and back-filled via capillary action via in-built filament for the self-made 1–2 μm micropipettes. Pressure supplied to the micropipettes is controlled via an Elveflow OB1 Mk3+ microfluidic flow regulator (Elveflow).

Micro-positioning of the micropipettes was accomplished through a three-axis micro-positioner system, with further information provided in Supplementary Section 1.

### Particle velocimetry

Particle tracking is performed to visualize and quantify fluid velocity within the confined fluid envelope during transit from injection to aspiration micropipette. Gold nanoparticles (diameter 80 nm) are chosen as probe particles. iSCAT microscopy is performed at 10,000 fps.

Further details on twin-micropipette geometry are provided in Supplementary Section 2.

### Cell culture

COS-7 cells (DSMZ, catalog no. ACC 60) were grown in DMEM (Gibco, Invitrogen) supplemented with 10% fetal calf serum (Life Technologies) in a humidified atmosphere at 37 °C and 5% $CO_2$. For measurement, cells were plated onto 50 mm glass-bottomed dishes (MatTek, 30 mm growth area) and grown to 70–80% confluency. Before measurement, each dish was rinsed three times with Dulbecco's PBS (DPBS) (Gibco, Invitrogen). Imaging was performed in Leibovitz's L-15 medium (3 ml, Gibco, Invitrogen) at room temperature (22 °C).

### Surface patterning demonstration

Carboxylate-modified fluorescent microspheres, diameter 200 nm, with 488 nm fluorescence (F8811) were purchased from Invitrogen with a concentration of $4.5 \times 10^{12}$ beads per ml. The beads were diluted 1:100 and 1:1,000 in DPBS.

### FRAP and μkiss experiments

**Supported lipid bilayer preparation.** All lipids (Avanti Polar Lipids) were purchased from Merck: DOPC (catalog no. 850375C), DOPS (catalog no. 840035C) and GM1 (catalog no. 860065P). Working solutions of each lipid species were prepared as the following: DOPC and DOPS were reconstituted in chloroform to obtain concentrations of 10 and 1 mg ml$^{-1}$, respectively. GM1 was reconstituted in a mixture of 80% chloroform and 20% methanol to a working concentration of 1 mg ml$^{-1}$. For the small unilamellar vesicle (SUV) mix, 95% DOPC, 5% DOPS and 1% GM1 were used, the solvent was removed by blow-drying followed by incubation in a desiccation chamber. Milli-Q water was added to obtain a lipid concentration of 1 mg ml$^{-1}$, followed by vortexing the mixture and transferring it to a 15 ml falcon tube. After 50 min of sonicating in a tip sonicator, the lipid mix was centrifuged for 20 min at 4 °C and 16,000g. The supernatant was collected and stored at 4 °C until usage. For supported lipid bilayer formation and imaging, coverslips were cleaned in a series of sonication steps in detergent (2% Hellmanex), water and alcohol (ethanol and isopropanol) followed by plasma cleaning. The SUV mix was diluted 1:4 in Milli-Q water and pipetted onto the cleaned coverslip. Bilayer formation was induced by adding 100 mM calcium chloride in a concentration of 10% of the SUV mix volume. The bilayer was then washed three times with Milli-Q water and three times with DPBS. Imaging was performed in fresh DPBS solution.

**FRAP.** To perform GM1 FRAP experiments, we used CTxB with an Alexa Fluor 488 Conjugate (Invitrogen, catalog no. C34775), which was reconstituted to a stock concentration of 1 mg ml$^{-1}$. We incubate supported lipid bilayers in 400 μl DPBS containing 5 μg of CTxB-AF488 for 2 min, followed by removal of the staining solution and three DPBS washing steps. For FRAP experiments on live COS-7, 100,000 cells were seeded in 50 mm glass-bottom dishes (MatTek, 30 mm growth area) on the day before the experiment.

For imaging, a supercontinuum white light laser set to a center wavelength of 475 nm with a bandwidth of 50 nm was focused through an oil-immersion objective (Olympus, NA 1.4) to a bleaching spot with 1 μm diameter. Bleaching was performed for 1 s followed by recording of the fluorescence recovery.

**μkiss.** CTxB-AF488 was diluted to a working concentration of 0.1 mg ml$^{-1}$ for labeling. Labeling was performed with home-made micropipettes with spot sizes of 1 μm. Imaging of the label deposition and subsequent diffusion was completed with confocal fluorescence microscopy.

### Microtubules

Demecolcine was purchased from Sigma-Aldrich (catalog no. D1925) and used in a concentration of 27 μM. Tf-AF647 was added to the diluted

demecolcine solution to visualize the flow envelope when targeting live COS-7 cells transfected with the pmEGFP-a-tubulin-IRES-puro2b plasmid.

## Nanoplastics

Carboxylate-modified fluorescent microspheres, diameter 200 nm, with 488 nm fluorescence (F8811) were purchased from Invitrogen with a concentration of $4.5 \times 10^{12}$ beads per ml. The beads were diluted in DPBS to a concentration of $4.5 \times 10^{7}$ beads per ml.

## VLPs

The COVID-19 vaccine (Ad26.COV2-S (recombinant)) from Johnson & Johnson was used as a source for adenovirus-like particles encoding the SARS-CoV-2 spike glycoprotein. To fluorescently label the adenovirus-like particles, the buffer was stepwise replaced by borate buffer by using Proteus X-spinner 2.5 Ultrafiltration Concentrator Columns with a molecular weight cutoff of 100 kDa (Serva, catalog no. 42234.01) in five subsequent centrifugation steps (each 20 min, 4 °C, 25,000$g$). Then, an Atto488 NHS ester (Sigma-Aldrich, catalog no. 41698) was reacted in a 100-fold molar excess with the adenovirus-like particles in borate buffer for 1 h at room temperature. To remove non-bound fluorescent dye, fresh Proteus X-spinner 2.5 Ultrafiltration Concentrator Columns were used in five subsequent centrifugation steps as described previously. Purified adenovirus-like particles-Atto488 were diluted using the original vaccine supernatant recovered from the initial centrifugation steps. The concentration of the adenovirus-like particles in the vaccine as provided by the manufacturer is given as 8.92 $\log_{10}$ infectious units per 0.5 ml. Fluorescently labeled adenovirus-like particles were diluted in the vaccine supernatant to a final maximum working concentration of $3 \times 10^{7} – 3 \times 10^{8}$, assuming 100% recovery in the filtration steps.

## Clathrin fluorescent labeling and TfR stimulation

To fluorescently label overexpressed clathrin light chain, cells transfected with pSems-ClathrinLC-Halotag plasmid were stained with a HaloTag-OregonGreen ligand (Promega, catalog no. G2802) according to the manufacturer's protocol.

Human Tf-Alexa Fluor488 (Jackson ImmunoResearch Laboratories Inc., catalog no. 009-540-050) and Human Tf-Alexa Fluor647 (Jackson ImmunoResearch Laboratories Inc., catalog no. JIM-009-600-050) have been reconstituted to a stock concentration of 2 mg ml$^{-1}$ and used in a working concentration of 0.2 mg ml$^{-1}$ throughout all experiments.

## Actin

Latrunculin-A was purchased from Sigma-Aldrich (catalog no. L5163) and was used in a concentration of 20 μM. Tf-AF647 was added to the Latrunculin-A solution to visualize the flow envelope when targeting live COS-7 cells transfected with the mEmerald-Lifeact-7 plasmid.

## Statistics and reproducibility

All data presented in this paper are a fair representation of experiments performed. We obtained similar results for all experiments, with low statistical variation. All measurements have been repeated a number of independent times, and we indicate this within each figure via the parameter $n$. All parameters that result from a statistical ensemble are expressed as the sample mean ± their standard deviation. In Fig. 1e, particle tracking was performed on 200 nanoparticles from one single experimental run, and all trajectories are shown in the figure panel. This measurement was repeated separately three times using different micropipettes ($n = 3$). In Fig. 2, both FRAP and μkiss were separately performed on the same bilayer membrane for each measurement. Both bilayer membranes were produced from the same sample preparation. Each experiment was ran for a total of 30 measurements each ($n = 30$). Figure 2c presents data from one measurement, and Fig. 2e presents all

30 measured diffusion constants from both measurement modalities ($n = 30$). In Fig. 3b, punctual plasma membrane labeling was performed a total of eight times on different cells from the same culture sample ($n = 8$). The analogous FRAP measurement was also performed on a total of eight different cells from the same cell culture sample ($n = 8$). In Fig. 3e, demecolcine μkiss delivery was performed once on a single cell and repeated on a different cell for a total of five cells ($n = 5$). In Fig. 3j, nanoplastic particles were μkiss-delivered to a single COS-7 cell and repeated on a different cell for a total of three cells ($n = 3$). In Fig. 3k, VLPs were μkiss-delivered to a single COS-7 cell and repeated on a different cell for a total of three cells ($n = 3$). In Fig. 4, Tf-AF647 was μkiss-delivered to a single COS-7 and repeated on a different cell for a total of 13 cells ($n = 13$). In Fig. 5, Lat-A was μkiss-delivered to a single COS-7 and repeated twice on a different cell ($n = 2$).

## Reporting summary

Further information on research design is available in the Nature Portfolio Reporting Summary linked to this article.

## Data availability

Considering the large size of the individual raw videos, the datasets are available from the corresponding author on request. Requests will be answered within 3 weeks. Source data relating to Fig. 1 are available from Zenodo at https://doi.org/10.5281/zenodo.10419371.

## Code availability

Data capture was performed using proprietary software. Widefield iSCAT data acquisition was performed using the commercial camera control software of the camera manufacturer (PCC, Vision Research Inc.). Confocal microscopy data were acquired using Zeiss Zen v.3.0 (Carl Zeiss AG). Interpretation of data (that is, image background subtraction, Gaussian function fitting) was performed using MATLAB 2022a, using standard in-built functions. Code relating to the analysis of Fig. 4 is available at Zenodo (https://doi.org/10.5281/zenodo.10419371). Modeling of μkiss-labeled punctual diffusion was performed using the MATLAB code of Röding et al.[61], discussed further in Supplementary Section 6.

## References

61. Röding, M., Lacroix, L., Krona, A., Gebäck, T. & Lorén, N. A highly accurate pixel-based FRAP model based on spectral-domain numerical methods. *Biophys. J.* **116**, 1348–1361 (2019).

## Acknowledgements

We thank S. Ihloff for help with cell preparation matters and E. Butzen for SEM imaging. We thank D. Albrecht for critical reading of the manuscript and advice on biological matters. We additionally thank J. Lühr for her efforts in establishing pilot measurements with micropipettes. We also thank K. Almahayni for reading the manuscript as well as providing general support during the project. We especially thank B. Fabry for providing insightful discussions. This work was financed by the Max Planck Society. L.M. acknowledges funding from the Else Kröner Fresenius Stiftung (grant no. 2020_EKEA.91).

## Author contributions

C.H. and R.W.T. performed all measurements and conceived of the concept. L.M. advised on experimental demonstrations and interpretation of results. A.S. advised on interpretation of results. V.S. supervised the project and also advised on interpretation of results. C.H., R.W.T., L.M. and V.S. wrote the paper. All authors commented on the paper.

## Funding

## Competing interests

The authors declare no competing interests.

## Additional information

**Correspondence and requests for materials** should be addressed to Vahid Sandoghdar.

# Reporting Summary

## Statistics

For all statistical analyses, confirm that the following items are present in the figure legend, table legend, main text, or Methods section.

| n/a | Confirmed | |
|---|---|---|
| ☐ | ☒ | The exact sample size (*n*) for each experimental group/condition, given as a discrete number and unit of measurement |
| ☐ | ☒ | A statement on whether measurements were taken from distinct samples or whether the same sample was measured repeatedly |
| ☒ | ☐ | The statistical test(s) used AND whether they are one- or two-sided *Only common tests should be described solely by name; describe more complex techniques in the Methods section.* |
| ☒ | ☐ | A description of all covariates tested |
| ☒ | ☐ | A description of any assumptions or corrections, such as tests of normality and adjustment for multiple comparisons |
| ☐ | ☒ | A full description of the statistical parameters including central tendency (e.g. means) or other basic estimates (e.g. regression coefficient) AND variation (e.g. standard deviation) or associated estimates of uncertainty (e.g. confidence intervals) |
| ☒ | ☐ | For null hypothesis testing, the test statistic (e.g. *F*, *t*, *r*) with confidence intervals, effect sizes, degrees of freedom and *P* value noted *Give P values as exact values whenever suitable.* |
| ☒ | ☐ | For Bayesian analysis, information on the choice of priors and Markov chain Monte Carlo settings |
| ☒ | ☐ | For hierarchical and complex designs, identification of the appropriate level for tests and full reporting of outcomes |
| ☐ | ☒ | Estimates of effect sizes (e.g. Cohen's *d*, Pearson's *r*), indicating how they were calculated |

*Our web collection on statistics for biologists contains articles on many of the points above.*

## Software and code

Policy information about availability of computer code

| | |
|---|---|
| Data collection | Zeiss Zen 3.0 was used for recording confocal fluorescence and confocal iSCAT data. Phantom Camera Control (PCC) was used for recording highspeed single particle tracking data via iSCAT microscopy. COMSOL Multiphysics 6.0 was used for numerical simulations. |
| Data analysis | Data analysis comprises image processing, parameter extraction and also fitting of functions to data distributions (i.e. fitting of a 2D Gaussian). This was performed using MATLAB 2022a using standard in-built functions. |

For manuscripts utilizing custom algorithms or software that are central to the research but not yet described in published literature, software must be made available to editors and reviewers. We strongly encourage code deposition in a community repository (e.g. GitHub). See the Nature Portfolio guidelines for submitting code & software for further information.

## Data

Policy information about availability of data

All manuscripts must include a data availability statement. This statement should provide the following information, where applicable:
- Accession codes, unique identifiers, or web links for publicly available datasets
- A description of any restrictions on data availability
- For clinical datasets or third party data, please ensure that the statement adheres to our policy

Considering the large size of the individual raw videos, the datasets are available from the corresponding author on request. Requests will be answered within three

weeks. Source data are provided with this paper. Source data relating to Figure 1 is available via https://doi.org/10.5281/zenodo.10419371.

# Human research participants

Policy information about studies involving human research participants and Sex and Gender in Research.

| | |
|---|---|
| Reporting on sex and gender | No human participants were involved in this study. |
| Population characteristics | Not applicable |
| Recruitment | Not applicable |
| Ethics oversight | Not applicable |

Note that full information on the approval of the study protocol must also be provided in the manuscript.

# Field-specific reporting

Please select the one below that is the best fit for your research. If you are not sure, read the appropriate sections before making your selection.

☒ Life sciences　　　☐ Behavioural & social sciences　　　☐ Ecological, evolutionary & environmental sciences

For a reference copy of the document with all sections, see nature.com/documents/nr-reporting-summary-flat.pdf

# Life sciences study design

All studies must disclose on these points even when the disclosure is negative.

| | |
|---|---|
| Sample size | In our study, we present our technology on a number of representative experimental scenarios, that did not require specific sample sizes. No claims were made regarding biological significance. All experiments were at least performed multiple times, each on different individual cells. In the case of diffusion studies on SLBs, FRAP and µkiss brushing were performed 30 times on the same membrane. To avoid variations between different SLB preparations, the same preparation material was used to form a bilayer for both the FRAP and µkiss measurements. |
| Data exclusions | Representative data is shown for each experiment. No data was excluded. |
| Replication | All data presented in this manuscript is a fair representation of experiments performed. All measurements have been repeated several times with different rounds of sample preparation. We obtained similar results for all experiments, with low statistical variation. In Fig. 1e, particle tracking was performed on 200 nanoparticles from one single experimental run, and all trajectories are shown in the figure panel. This measurement was repeated separately three times using different micropipettes (n = 3). In Fig. 2, both FRAP and µkiss were separately performed on the same bilayer membrane for each measurement. Both bilayer membranes were produced from the same sample preparation. Each experiment was ran for a total of 30 measurements each (n = 30). Fig. 2c present data from one measurement, and Fig. 2e presents all 30 measured diffusion constants from both measurement modalities (n = 30). In Fig. 3b, punctual plasma membrane labeling was performed a total of 25 times on different cells from the same culture sample (n = 25). The analogous FRAP measurement was also performed on a total of eight different cells from the same cell culture sample (n = 8). In Fig. 3e, demecolcine µkiss-delivery was performed once on a single cell, and repeated on a different cell for a total of five cells (n = 5). In Fig. 3j, nanoplastic particles were µkiss-delivered to a single COS-7 cell, and repeated on a different cell for a total of three cells (n = 3). In Fig. 3k, VLPs were µkiss-delivered to a single COS-7 cell, and repeated on a different cell for a total of three cells (n = 3). In Fig. 4, Tf-AF647 was µkiss-delivered to a single COS-7, and repeated on a different cell for a total of 13 cells (n = 13). In Fig. 5, Lat-A was µkiss-delivered to a single COS-7, and repeated on a different cell for a total of two times (n = 2). |
| Randomization | Randomization was not required to avoid bias during data analysis. |
| Blinding | Not required; all measurements of the same kind were analysed using the same exact code without modification. In this regard, the analysis is blind. |

# Reporting for specific materials, systems and methods

We require information from authors about some types of materials, experimental systems and methods used in many studies. Here, indicate whether each material, system or method listed is relevant to your study. If you are not sure if a list item applies to your research, read the appropriate section before selecting a response.

## Materials & experimental systems

| n/a | Involved in the study |
|-----|----------------------|
| ☒ ☐ | Antibodies |
| ☐ ☒ | Eukaryotic cell lines |
| ☒ ☐ | Palaeontology and archaeology |
| ☒ ☐ | Animals and other organisms |
| ☒ ☐ | Clinical data |
| ☒ ☐ | Dual use research of concern |

## Methods

| n/a | Involved in the study |
|-----|----------------------|
| ☒ ☐ | ChIP-seq |
| ☒ ☐ | Flow cytometry |
| ☒ ☐ | MRI-based neuroimaging |

## Eukaryotic cell lines

Policy information about cell lines and Sex and Gender in Research

| | |
|---|---|
| Cell line source(s) | COS-7 cells were obtained from the DSMZ (German Collection of Microorganisms and Cell Cultures GmbH). |
| Authentication | Cells were routinely validated through morphology. As the cells were obtained commercially, no further authentication was performed. |
| Mycoplasma contamination | Cells were routinely tested negative for mycoplasma contamination. |
| Commonly misidentified lines (See ICLAC register) | No commonly misidentified cell lines were used in our study. |

