## [Peer Review File · Nature Methods]

Peer Review Information

Manuscript Title: A paintbrush for delivery of nanoparticles and small molecules to live cells with micrometer spatial and millisecond temporal control

Corresponding author name(s): Vahid Sandoghdar

Editorial Notes: None

Reviewer Comments & Decisions:

Decision Letter, initial version:

Dear Vahid,

Please let me begin by apologizing for the slow speed of this review process. You had two reviews in for some time, but they gave opposing views, and we wanted to wait for the third to get a more balanced picture. Thank you for your patience.

Your Article, "A paintbrush for delivery of nanoparticles and small molecules to live cells with micrometer spatial and millisecond temporal control", has now been seen by four reviewers (two cosubmitted a review but are listed separately, so you have three reviews total). As you will see from their comments below, although the reviewers find your work of considerable potential interest, they have raised a number of concerns. We are interested in the possibility of publishing your paper in Nature Methods, but would like to consider your response to these concerns before we reach a final decision on publication. We therefore invite you to revise your manuscript to address these concerns.

We think most of the technical questions and calls for characterizations of robustness are fair and should hopefully be straightforward to address. Our overarching concern is whether the method truly enables new types of delivery (as raised by refs 1 and 2). We think this could be addressed in either of two ways, either direct experimental comparison on the same task or by showing another demonstration that clearly cannot be achieved with alternative technologies. We leave the choice up to you, but I am happy to discuss.

[Redacted] This URL links to your confidential home page and associated information about manuscripts you may have submitted, or that you are reviewing for us. If you wish to forward this email to co-authors, please delete the link to your homepage.

We hope to receive your revised paper within three months. If you cannot send it within this time, please let us know. In this event, we will still be happy to reconsider your paper at a later date so long as nothing similar has been accepted for publication at Nature Methods or published elsewhere.

OPEN SCIENCE REQUIREMENTS

REPORTING SUMMARY AND EDITORIAL POLICY CHECKLISTS

Please note that these forms are dynamic ‘smart pdfs’ and must therefore be downloaded and completed in Adobe Reader. We will then flatten them for ease of use by the reviewers. If you would like to reference the guidance text as you complete the template, please access these flattened versions at <http://www.nature.com/authors/policies/availability.html>.

DATA AVAILABILITY

All novel DNA and RNA sequencing data, protein sequences, genetic polymorphisms, linked genotype and phenotype data, gene expression data, macromolecular structures, and proteomics data must be deposited in a publicly accessible database, and accession codes and associated hyperlinks must be provided in the “Data Availability” section.

Please include a “Data availability” subsection in the Online Methods. This section should inform readers about the availability of the data used to support the conclusions of your study, including accession codes to public repositories, references to source data that may be published alongside the paper, unique identifiers such as URLs to data repository entries, or data set DOIs, and any other statement about data availability. At a minimum, you should include the following statement: “The data that support the findings of this study are available from the corresponding author upon request”, describing which data is available upon request and mentioning any restrictions on availability. If DOIs are provided, please include these in the Reference list (authors, title, publisher (repository name), identifier, year). For more guidance on how to write this section please see: <http://www.nature.com/authors/policies/data/data-availability-statements-data-citations.pdf>

CODE AVAILABILITY

Please include a “Code Availability” subsection in the Online Methods which details how your custom code is made available. Only in rare cases (where code is not central to the main conclusions of the paper) is the statement “available upon request” allowed (and reasons should be specified).

For more information on our code sharing policy and requirements, please see: <https://www.nature.com/nature-research/editorial-policies/reporting-standards#availability-of-computer-code>

MATERIALS AVAILABILITY

SUPPLEMENTARY PROTOCOL

To help facilitate reproducibility and uptake of your method, we ask you to prepare a step-by-step Supplementary Protocol for the method described in this paper. We [encourage authors to share their step-by-step experimental protocols](https://www.nature.com/nature-research/editorial-policies/reporting-standards#protocols) on a protocol sharing platform of their choice and report the protocol DOI in the reference list. Nature Portfolio's Protocol Exchange is a free-to-use and open resource for protocols; protocols deposited in Protocol Exchange are citable and can be linked from the published article. More details can found at www.nature.com/protocolexchange/about.

ORCID

Sincerely,
Rita

Rita Strack, Ph.D.
Senior Editor
Nature Methods

Reviewers' Comments:

Reviewer #1:

Remarks to the Author:

This manuscript describes a hand-made paintbrush device (MicroKiss) by using two closing sharp pipettes to control the laminar flow envelop for delivery of nanoparticles and small molecules onto live cells with micrometer spatial and sub-second temporal control ability. Although very careful calculation/simulation had been illustrated to verify the capability of this hand-made system for delivering molecules/particles onto a local region on a single cell level with about 10 μm spatial and 0.13 s temporal resolution, however, the uncertainty of the device fabrication and tedious arrangement/operation process can not be considered more advanced than microfluidic/nanofluidic systems. Those demonstrated operations on single cells can also be easily done by using many other systems (AFM, Micro/Nanofluidic systems, Micro-optical electrical systems, etc..) with even better temporal and spatial control. I do not feel the proposed device provides a novel and better operation on single cell delivery. Some more detailed suggestions are listed below for author to consider:

1. How to control and keep the accurate size of and distance between two apertures? What will be the average variation of the size and distance of the apertures for each fabricated pair device? It seems the control and maintenance process were very complex/tedious but not mentioned in-detail in the manuscript. How sensitive the size or distance change of the two apertures will affect the Qratio? There is also no quantitatively statistic data about the repeatability and size, distance control among apertures and between tip and cells.
2. How The compliance of the long slender pipette affects the microKiss performance? Will the distance change during the operation at different flow speed/rate by the deformation of the long and slender pipette tips from the force disturbance of fluid flow or vibration/acoustic/Brownian motion from the environment?
3. Regarding the experimental verification about the diffusion issue, current experiment was conducted by using 80 nm Au nanoparticles. However, not only the density are 19 times higher than water but also the diffusivity is 200 times lower than molecular dyes, posing large deviation from the real applications. The authors are recommended to use molecular dye rather than 80 nm Au nanoparticles for the validation of issue of molecular diffusion.
4. The millisecond temporal control is over-claimed for the microKiss device since the demonstrated time repose is only 130 ms, merely sub-second range, quit slow action related to protein response in ms-microsecond range on cell membrane.
5. Even though the delivery speed for one type chemical can be controlled on and off in 130ms, however, to delivery different chemicals sequentially cannot be controlled very easily by using this microKiss system when compared to the micro/nano fluidic systems, and the later can provide much more precise multiplexing chemicals in better temporal position, and dose control. I do not feel this technic is more advanced than most of the micro/nano fluidics systems for controlling localized chemicals delivery onto single cells in temporal and spatial resolution.

A. Summary of the key results:

Clear and OK

B. Originality and significance: if not novel, please include reference

Not novel, when compared to micro/nanofluidic systems with more precise spatial and much better temporal control plus multiplexing capabilities.

C. Data & methodology: validity of approach, quality of data, quality of presentation

No statistic data on fabrication, arrangement, operation errors of the device. No information about the disturbance from fluidic flow, brownian motion, acoustic, vibration effects to the operation precision of the device.

D. Appropriate use of statistics and treatment of uncertainties

Basically those are ok for the demonstrated 5 application cases, but not for device itself.

E. Conclusions: robustness, validity, reliability

Overclaimed on the temporal resolution and utility.

F. Suggested improvements: experiments, data for possible revision

As stated above.

G. References: appropriate credit to previous work?

Should compared more with the results from micro/nanofluidic systems and AFM operation (especially Dip pen lithography)

H. Clarity and context: lucidity of abstract/summary, appropriateness of abstract, introduction and conclusions

Basically those are ok except mentioned above.

Reviewer #2:

Remarks to the Author:

In this manuscript the authors report a dual micropipette-based method termed μ Kiss that allows for localized delivery of biomolecules to the surface of lipid membranes and cells. The principle of operation involves fluid injection using one micropipette and aspiration using the other to create a fluid jet envelope where the molecules to be delivered are confined. The molecules are painted on the surface of the target membrane when this fluid jet is brought in close vicinity. The authors first characterize the fluidic performance of the system using numerical simulation and controlled experiments. Then they demonstrate the use of this technique to determine the transport properties of a model membrane by locally painting it with a fluorescent molecule. The authors present a comparison of their method with FRAP and find good agreement. The authors then study receptor diffusion in sub-cellular regions using this method which is challenging using conventional techniques. The authors also demonstrate the utility of this method in labeling cells with single nano/viral particles and local physio-chemical small molecule disruptors. Finally, the authors demonstrate the potential use of this method in studying cell-cell interactions by locally disrupting the cytoskeletal structure of one cell and evaluating the impact on neighboring cells.

This study is well executed and reported. The writing is clear and conclusions are appropriate. The work is original. The method developed does have unique and novel utility, but its significance could be better highlighted. Major impactful applications that appeal to a larger scientific community should be further discussed. References are appropriate but a body of literature constituting single cell intracellular delivery should be discussed in context of the current work (details below). Conclusions are robust and reliable. The statistical design of experiments and analysis are good for several experiments but could be better represented (details below).

Overall, revisions are suggested to improve the manuscript so that it can be considered for publication in this journal. Below are some specific questions and suggestions:

1. The authors mention that single cell delivery in populous cultures is precluded when using AFM/Micropipette systems (line 70-72). However, single cell viral and nanoparticle stamping has been demonstrated previously as discussed by the authors in this manuscript. Additionally, AFM and micropipette-based methods that allow for single cell intracellular delivery in cell populations have also been reported previously. As such, single cell delivery is not unique to the μ Kiss method. Would it be more appropriate to claim that sub-cellular localization is the unique feature/novelty of this method? Can this method also be used for confined intracellular delivery or is it limited to superficial delivery? If there are such limitations, then these should also be discussed. In general, the introduction should cover some the reports listed below in the context of the current study.
 - a. Kang et. al., Nanofountain Probe Electroporation (NFP-E) of Single Cells, Nano Letters, 2013
 - b. Shi et. al., Electrochemical Single-Cell Protein Therapeutics Using a Double-Barrel Nanopipette, Angewandte Chemie, 2022
 - c. Yang et. al., Single-cell membrane drug delivery using porous pen nanodeposition, Nanoscale, 2022
 - d. Mukherjee et. al., Deep Learning-Assisted Automated Single Cell Electroporation Platform for Effective Genetic Manipulation of Hard-to-Transfect Cells, Small, 2022
2. The vicinity of the brush and the target membrane is controlled by focusing the two within the depth of focus of the objective and may have some variability. Can this positional variability introduce any significant variability in labeling area/dosage?
3. The time taken to establish a fully developed envelope from actuation is ~ 100 ms based on the experiments. Is the timescale to change the flow profile (1% threshold envelope size) from a larger to a smaller brush size similar?
4. A control experiment is required to establish the superior spatial and temporal confinement of the μ Kiss method. Can the authors perhaps draw a 1 to 1 comparison of the confinement of a cargo on a membrane using the μ Kiss with a delivery based on microinjection only (by turning off the aspiration)? This is important as it will highlight the novelty of this technique and unique use case scenarios such as those mentioned in the TfR-clathrin membrane dynamics study – spatially/temporally localized labelling to reveal unknown dynamics, multiple labeling on same cell to study interactions etc.
5. The live dead assay seems to be presented for one cell. A statistically meaningful population should be tested or if tested already, the data should be added to the manuscript.

6. Continuing from the previous comment, the authors have performed the different experiments on multiple cells. To highlight the full body of work to the readers they are encouraged to think if there are statistically meaningful ways of presenting the full datasets in the figures in addition to the snapshots.

Reviewer #3:

Remarks to the Author:

Here Holler, Taylor and colleagues report the development and use of a microfluidic delivery device to spray very small volumes (Femtoliters) of particle-suspensions and/or drugs to the surface of individual cells while under observation by the light microscope. They use a pair of microneedles coupled to a micromanipulator to “apply” their reagent of interest to the cell surface while simultaneously removing excess reagent with a suction micropipette. The result is a system that can “paint” a region on the surface of a cell with relatively well-defined volumes of a variety of molecules. This paper is well-written, provides a comprehensive description of the technique including extensive controls (effect of shear-stress on cell surface vs induction of cell death, comparison of multiple cellular responses to changes in lipid bilayer composition, cytoskeleton dynamics, and endocytosis. This study will be of great interest to the readership of Nature Methods.

The manuscript requires relatively few changes. This reviewer does have a question on the organization of references – perhaps it is a journal style, but the references are neither numbered sequentially, nor alphabetized. Instead, they are inserted randomly, based on order of usage. This is awkward to use, as the reader must scan the entire list to find a particular reference. Numbered references or alphabetized references should be used instead.

One reference to the literature that should be included is O’Connell, Warner and Wang, 2001 Current Biology 1:702–707. This study uses a pair of micropipets (delivery-suction) to release cytochalasin D onto the surface of dividing NRK cells to examine localized actin dynamics at the cell cortex. The current work represents an advance over the system described 20 years ago, and the present manuscript provides both a detailed methodological description of the technique and sufficient control experiments to warrant publication in nature Methods. However, it would be helpful to contrast this current study with the technique used by the Wang group.

Reviewer #4:

None

Author Rebuttal to Initial comments

We thank the three Referees each for their constructive comments and suggestions that we have endeavoured to incorporate into our manuscript, much to its benefit. We address the specific remarks in the following.

Referee 1

Although very careful calculation/simulation had been illustrated to verify the capability of this hand-made system for delivering molecules/particles onto a local region on a single cell level with about 10 μm spatial and 0.13 s temporal resolution, however, the uncertainty of the device fabrication and tedious arrangement/operation process can not be considered more advanced than microfluidic/nanofluidic systems. Those demonstrated operations on single cells can also be easily done by using many other systems (AFM, Micro/Nanofluidic systems, Micro-optical electrical systems, etc..) with even better temporal and spatial control. I do not feel the proposed device provides a novel and better operation on single cell delivery.

Our reply

Firstly, we thank Referee #1 for their thoughtful and critical assessment of our work. We politely disagree with the claim that our proposed solution neither provides a novel nor better reagent delivery solution to the single cell as compared to previously reported methods. Our novelty is the *in situ* delivery of reagents in the close *external* vicinity of the cellular membrane, which we now call the “near-field” of the cell. We do this in a contact-free fashion to regions as small as a few microns (well below 10 μm). The purpose is to study the interaction of the material on the extra-cellular space of the cell, whether that be diffusion, membrane translocation, or uptake. We have now specified this more clearly in the manuscript.

It is useful to recapitulate several unique advantages of our approach we have demonstrated:

- **Precise directed delivery:** We demonstrate membrane administration of reagents, down to 2 μm spatial resolution. The area upon which reagents are delivered is freely chosen by precise steering of the micropipettes. Reagents introduced into the open volume are confined within the flow envelope and any unbound reagents are removed from the open fluidic volume, preventing unwanted exposure, cross-contamination or medium fouling. Delivery can be executed in a time window as prompt as 35 ms [See revised **Supporting Information Section 5**].
- **Easy to use:** The majority of cell-based measurements are typically conducted using culture dishes or well-plates combined with optical microscopy. Our approach seamlessly integrates with these widely adopted methods, and serves as a simple augmentation. Our method can be performed on chosen individual cells *in situ* within their native context within the dish.
- **Wide reagent compatibility:** By using chiefly convection-based transport, reagent administration can be performed with virtually any material, independent of the material’s particular properties (size, charge, mass), and without need for calibration.

We find our performance is unmatched by any existing system, in particular:

- **Fluidic AFM (FluidFM):** Microfluidic AFM, known as Fluid FM, dispenses fluids through a small aperture ranging from 200nm to several microns. Typically, it is primarily employed for *intracellular* material delivery [doi 10.1002/admi.202001115] although in rare instances it’s also been used to deliver a single virus to the external membrane [doi 10.1021/nl3018109]. However, this approach has drawbacks, including contact between the cell and probe head. Moreover, such methods face a

fundamental issue when they aim to deliver a small number of nanoparticles or molecules: Reducing the number means reducing the concentration, i.e., the ratio of the material of choice and the carrying solvent. This in turn means that a large amount of liquid is delivered to transmit only a few particles of choice.

- **Dip Pen Nanolithography (DPN):** DPN employs a sharp nanoscale tip coated with a solvated reagent, which is then brought into contact with the target surface, facilitating reagent transfer through solvent diffusion. While DPN is effective in various applications, it is not commonly used with live cells due to its operation in air and incompatibility with aqueous media. The use of oil as a solvent can address this limitation but is incompatible with many reagents. Prolonged contact with the probe surface may also cause denaturation of biological reagents, a challenge shared with similar techniques like Porous Pen Nanodeposition (PPN). This method is not suitable for studying physiologically relevant processes of how a molecule or nanoparticle “finds its way” to the cell. Here, it is important to keep in mind that the cell surface contains many hair-like biopolymer and molecules (e.g., sugars) that would be perturbed by the introduction of a DPN.
- **Micro/Nanofluidic Substrates:** Micro/nanofluidic substrates are intricately crafted chips designed for precise fluid manipulation. However, they come with several drawbacks, including their challenging fabrication process, limited ease of use compared to standard culture dishes, and difficulties in integration with other systems or microscopy equipment. The enclosed nature of these substrates makes routine processing steps inconvenient, e.g., because their narrow channels are susceptible to blockages. Additionally, nanoscale channels introduce sample-wall interactions that can interfere with reagent and cell behaviour. Furthermore, their small dimensions pose challenges for microscopy and imaging. Delivery in such devices can be realized but at well-defined positions; real-time change of delivery location would require very sophisticated designs.
- **Micro-optical Electrical Mechanical Systems (MOEMS):** MOEMS - small devices combining micro-fabrication with optical, electrical and mechanical technologies, have not, to the best of our knowledge, shown the capability to steer reagent delivery precisely to microscale regions on the cell membrane. Owing to the shortcomings of these technologies that MOEMS integrate (i.e., optical tweezers, optogenetics, electrophoresis - many of which we address in the manuscript), we similarly do not see a case for MOEMS being able to provide a superior solution for directed reagent delivery to single cells.

After a more careful review of some of the suggestions made by the referee, we remain convinced that there is no technique other than ours that can claim to accomplish the same degree of confined and controlled reagent delivery and with our spatial and temporal resolution. This established the novelty of our approach.

We also would like to take this opportunity to gently correct a statement made by the referee: our resolution is not 10 μm , but is in fact better than 2 μm , as we show in **Figure 3**. The spatial resolution by which we can achieve our delivery is extremely useful when seeking to address fundamental questions in cell biology. Moreover, we note that in applications where a large envelope is used, i.e., in **Figure 4**, we show administration to the membrane of a single 200 nm virus particle. In this application, it was not necessary to confine the delivery more.

The Referee also raises concern about the uncertainty of fabrication and the tediousness of implementing our solution. We have now revised our supplementary material to emphasize that the fabrication is fairly simple. Furthermore, the uncertainties are small (<10%) and do not affect operation significantly. We

address this issue in more detail in response to **Remark #1**. Similarly, alignment of the tips is readily accomplished, and is far from being tedious. We also address this concern in more detail in response to **Remark #1**.

Referee 1

How to control and keep the accurate size of and distance between two apertures? What will be the average variation of the size and distance of the apertures for each fabricated pair device? It seems the control and maintenance process were very complex/tedious but not mentioned in-detail in the manuscript. How sensitive the size or distance change of the two apertures will affect the Qratio? There is also no quantitatively statistic data about the repeatability and size, distance control among apertures and between tip and cells.

Our reply

The referee raises valid practical concerns regarding the ease and reproducible operation of the device. We realize that many of these concerns could have been addressed had we provided greater detail of operation. We thank the referee for pointing out our oversight. We have now added four figures to **Section 1** of the **Supporting Information** to address the questions raised.

Firstly, we must clarify that operation of the micropipette-pair is actually quite simple. Equally, so is the control over the flow envelope. The dual micropipette system is quite forgiving to operate, and gives reproducible performance even against tip misalignment and inherent variations in fabrication from micropipette-to-micropipette. This we have verified through both experiment and numerical modeling. We tested that even for non-experienced users, it takes no more than a few minutes to establish the desired flow envelope and commence the experiment.

The pipettes vary in their aperture dimensions by less than 10%, which has a negligible effect on the geometry of the flow envelope [see **Revised Supporting Information, Figs. 6-8**]. Positioning and alignment of the pipette-pair to sub-microscale precision is readily accomplished by a pair of motorized micro-positioners while monitored in the microscope. The positioning units comprise a *common* platform that allows the pair to be moved in perfect tandem [see **Revised Supporting Information, Fig. 5**]. In addition, a second positioner on the platform independently aligns one micropipette with respect to other. The micropipettes are measured to be positionally stable to ca. 100 nm, while a misalignment of several microns is required to affect brush operation significantly [see **Revised Supporting Information, Fig. 8**]. It, thus, follows that the flow envelope is easily robust.

To assuage concerns of reproducibility of operation, we have also included in **Revised Supporting Information, Fig. 9**, a series of illustrations of how the brush can be steered to administer fluorescent beads to the coverslip in a series of consistent and well-defined patches. We show labeling of beads with sub-poissonian patch-to-patch variation in bead number.

Referee 1

How the compliance of the long slender pipette affects the microKiss performance? Will the distance change during the operation at different flow speed/rate by the deformation of the long and slender

pipette tips from the force disturbance of fluid flow or vibration/acoustic/Brownian motion from the environment?

Our reply

We appreciate the referee's concern, however, the micropipettes in our system are stable and the performance consistent. We observed no change in the ca. 100 ± 2 nm positional uncertainty of the micropipettes whether flow was actuated or not – which we present in **Revised Supporting Information, Fig. 7**. From this we conclude that effects of flow rate are negligible, especially given the low pressures with which we operate. Furthermore, we do not observe significant disturbance from acoustic or vibrational sources in the measurement environment.

Referee 1

Regarding the experimental verification about the diffusion issue, current experiment was conducted by using 80 nm Au nanoparticles. However, not only the density are 19 times higher than water but also the diffusivity is 200 times lower than molecular dyes, posing large deviation from the real applications. The authors are recommended to use molecular dye rather than 80 nm Au nanoparticles for the validation of issue of molecular diffusion.

Our reply

Again, we appreciate the referee's comment and excellent recommendation, which we have happily implemented in **Revised Supporting Information, Fig. 14**. We have chosen fluorescein as a small molecular dye (molecular Mass=332 Da, diffusivity $D=435 \mu\text{m}^2\text{s}^{-1}$ [doi 10.1021/jp207459k]), and validated flow with both experiment and numerical modeling. In summary, the demonstration with Au nanoparticles is representative. We note that the use of Au nanoparticles allowed us to present a uniquely high-speed and precise demonstration since we could fully track each nanoparticle.

Referee 1

The millisecond temporal control is over-claimed for the microKiss device since the demonstrated time repose is only 130 ms, merely sub-second range, quit slow action related to protein response in ms-microsecond range on cell membrane.

Our reply

We respectfully disagree that the temporal response is over-claimed. After all, a 130 nm particle is not considered to be a microparticle. Nevertheless, we appreciate the sentiment that a faster response time would strengthen the utility of the method. The previously stated temporal resolution was not at the limit of our performance. We have since optimised our apparatus and demonstrate a response time of 35 ± 2.5 ms in **Revised Supporting Information, Fig. 17**.

Referee 1

Even though the delivery speed for one type chemical can be controlled on and off in 130ms, however, to delivery different chemicals sequentially cannot be controlled very easily by using this microKiss system

when compared to the micro/nano fluidic systems, and the later can provide much more precise multiplexing chemicals in better temporal position, and dose control. I do not feel this technic is more advanced than most of the micro/nano fluidics systems for controlling localized chemicals delivery onto single cells in temporal and spatial resolution.

Our reply

The referee points out correctly that we cannot multiplex different reagents through our apparatus. This is an interesting feature for special applications, but it is neither the aim of our method nor of many of the other existing techniques that the referee listed in their opening statement. The majority of cell-based investigations are performed in an open dish and not with microfluidics. Our approach aims to cater to the large community of cell biologists to offer them contact-free confined microscale reagent delivery steered towards the extra-cellular matrix. As discussed in previous responses, this clarity and resolution of sub-cellular targeting has not been demonstrated using micro/nano fluidic systems. We emphasize again that confined delivery is a key desirable feature. In other words, it is important to avoid introducing the reagent to the buffer unless it has a very high chance of interacting with the cell membrane.

In closing, we are grateful to Referee 1 for raising many interesting questions. We believe by addressing these in our manuscript and Supplementary Information, we have improved the accessibility of our work to a broader community of scientists.

Referee 2

The authors mention that single cell delivery in populous cultures is precluded when using AFM/Micropipette systems (line 70-72). However, single cell viral and nanoparticle stamping has been demonstrated previously as discussed by the authors in this manuscript. Additionally, AFM and micropipette-based methods that allow for single cell intracellular delivery in cell populations have also been reported previously. As such, single cell delivery is not unique to the μ Kiss method. Would it be more appropriate to claim that sub-cellular localization is the unique feature/novelty of this method? Can this method also be used for confined intracellular delivery or is it limited to superficial delivery? If there are such limitations, then these should also be discussed. In general, the introduction should cover some the reports listed below in the context of the current study.

- a. Kang et. al., Nanofountain Probe Electroporation (NFP-E) of Single Cells, Nano Letters, 2013*
- b. Shi et. al., Electrochemical Single-Cell Protein Therapeutics Using a Double-Barrel Nanopipette, Angewandte Chemie, 2022*
- c. Yang et. al., Single-cell membrane drug delivery using porous pen nanodeposition, Nanoscale, 2022*
- d. Mukherjee et. al., Deep Learning-Assisted Automated Single Cell Electroporation Platform for Effective Genetic Manipulation of Hard-to-Transfect Cells, Small, 2022*

Our reply

We thank the referee for raising an important concern wherein it is apparent that we ought to articulate the uniqueness of our approach more exactly. We have reworded several aspects of the introduction to achieve so. In brief, our method is motivated to mimic the initial encounter and interaction of the reagent

with a cell, in a near to physiological fashion, with microscale resolution and prompt temporal control. This enables for quantitative investigation of biological processes that are initiated by the presence of external agents such as pathogens, ligands, or pharmaceutical molecules.

By being microscale in the region to which the confined targeting may be steered, very fine and selective probing of the cell at the level of its own anatomical elements can be performed. The unity of these aspects in one realization renders a powerful labeling performance that has not previously been reported. Furthermore, with regard to methods for intracellular injection, our explicit intention with μ kiss is to not put materials inside of the cell, rather to deliver them to the extra-cellular space we refer to as the 'near-field'. Of the reagents administered, what wants to go into the cell (or rather, what the cell will admit inside), will go inside. Often this process is itself the subject of study.

We have amended several sentences in the introduction to better clarify this aspect, as well as incorporating the reports suggested by the referee into the discussion. Furthermore, we also appreciate the referee's recommendation that we should provide a clearer discussion on the scope of μ kiss application, and its use for investigating intracellular translocation, but not as a primary means of intracellular injection.

Referee 2

The vicinity of the brush and the target membrane is controlled by focusing the two within the depth of focus of the objective and may have some variability. Can this positional variability introduce any significant variability in labeling area/dosage?

Our reply

The referee raises an interesting and important question. The positional uncertainty of our apparatus does not introduce a significant variation in labeling performance, and if an experimenter was especially concerned about the exact labeling area, it is always possible to calibrate the brush position in the axial direction on a few test cells until a satisfactory profile is obtained, and then start the experiment.

To verify that our apparatus does not introduce a significant variation in labeling performance, we have performed numerical simulations to visualize how variations in the axial position of the flow envelope, with respect to the target surface, will affect the labeling area, shown in **Figure I** below. We have chosen flow parameters consistent with those used in experiments, and examined axial planes in the range of $\pm 1 \mu\text{m}$ to generously mimic the potential variability in the axial position resulting from uncertainty in focusing. **Figure 1a** present a view of the μ kiss envelope intersecting the target surface, mimicking μ kiss operation. **Figure 1b** presents the μ kiss-labeled areas on the surface corresponding to differing axial positions of the envelope with respect to the surface, wherein it can be seen that no significant change to the labeled area, nor its dosage, results.

Figure 1: Variations in labeling area as a result of variations in axial position. a, COMSOL simulation of the μkiss from a micropipette-pair with $Q_{inj} = 0.16 \text{ nLs}^{-1}$ and $Q_{ratio} = 0.3$. Shown is an isosurface of the reagent at its stock concentration. b, Labeled patches that result from the indicated displacement of the micropipette-pair along the axial z-direction.

We also take this opportunity to add that the amount and rate of material administered is a frequently encountered challenge for all aperture-assisted delivery methods. To know this information, one requires an indicator and means of detection to allow for feedback and adjustment. Our method accomplishes this by imaging and monitoring the reagent delivery and using fine micro-positioners to provide adjustments. If the results directly observed are not satisfactory, e.g., if too little or too much is delivered, the delivery is iterated either on the same target cell or a neighboring one until the desired results are obtained.

Referee 2

The time taken to establish a fully developed envelope from actuation is $\sim 100 \text{ ms}$ based on the experiments. Is the timescale to change the flow profile (1% threshold envelope size) from a larger to a smaller brush size similar?

Our reply

Yes. We measured the response time for establishing the flow envelope from half-to-full extent (repeated over many cycles) to be approximately the same time as establishing the full extent of the brush from complete extinction. We include these findings within **Revised Supporting Information, Fig. 17**. This figure supplants the previous figure of the envelope establishment time, as we have sought to improve the

temporal response of our apparatus, and can now demonstrate a rise time of 35 ± 2.5 ms to establish the flow envelope.

Referee 2

A control experiment is required to establish the superior spatial and temporal confinement of the μ Kiss method. Can the authors perhaps draw a 1 to 1 comparison of the confinement of a cargo on a membrane using the μ Kiss with a delivery based on microinjection only (by turning off the aspiration)? This is important as it will highlight the novelty of this technique and unique use case scenarios such as those mentioned in the TjR-clathrin membrane dynamics study – spatially/temporally localized labelling to reveal unknown dynamics, multiple labeling on same cell to study interactions etc.

Our reply

We are happy to provide such a measurement and thank the referee for this suggestion. In **Fig. 3**, we demonstrated an archetypical measurement typical of membrane investigations, comprising the labeling of lipids in the plasma membrane of the live cell. We targeted GM1 lipids using a 1mM CTxB-AF488 solution. Through μ kiss delivery, we tagged GM1 lipids within a compact spot of 2 μ m over the course of one second (with labeling sufficient to perform subsequent mobility studies).

We have now performed a control experiment for this demonstration, using as suggested, a single micropipette under injection-only operation, which we have included as **Revised Supporting Information Fig. 24 (Section 11)**. In this measurement, using a micropipette with a competitively small 500 nm aperture, operated under minimal flow rates, one cannot achieve the localized and prompt labeling performance of μ kiss. This is because (1) even from the smallest apertures, injected material is spatially dispersed out in a wide jet, precluding localized microscale confinement and spread into a wider volume. We visualize the extent of the jet by performing high-speed particle tracking on the micropipette (**Revised Supporting Information Fig. 24**) to illustrate this point. (2) The spatial dispersion of the injected jet dilutes the effective concentration of the label at the membrane, resulting in longer labeling times to achieve similar labeling densities as μ kiss.

Referee 2

The live dead assay seems to be presented for one cell. A statistically meaningful population should be tested or if tested already, the data should be added to the manuscript.

Our reply

We concur with the referee that greater statistics should be provided for the LIVE/DEAD assay. We have amended **Revised Supporting Information Fig. 16** to include DEAD signal monitoring from a larger population of 10 cells over a longer period of 30 mins, and included both positive and negative DEAD signals for reference. Our results show all of the ten cells retain their LIVE signal and show no onset of DEAD signal, affirming the viability of our approach. We have amended the main text to include this information.

Referee 2

Continuing from the previous comment, the authors have performed the different experiments on multiple cells. To highlight the full body of work to the readers they are encouraged to think if there are statistically meaningful ways of presenting the full datasets in the figures in addition to the snapshots.

Our reply

The referee makes a very reasonable appeal for completeness of information. The response of cells is typically highly heterogeneous so that a quantitative assessment of a given phenomenon requires repeated measurements and statistical analysis. Quantities such as diffusion constant, transport speed, sticking probability, etc can be examined in dedicated measurements. We have indeed done that for a few cases such as lipid diffusion in SLBs (Fig. 2) and for cells as presented in Fig. 3b of the manuscript and Section 6 of the SI. We have performed all our experiments on several cell samples to assure their robustness, but our aim has not been to study any particular biological question or parameter in this work. Thus, we could not find any convenient way to express any statistical data. In our work, the experiments performed on cells are to demonstrate the applicability and potential of the method for cell biological studies.

Referee #3

This reviewer does have a question on the organization of references – perhaps it is a journal style, but the references are neither numbered sequentially, nor alphabetized. Instead, they are inserted randomly, based on order of usage. This is awkward to use, as the reader must scan the entire list to find a particular reference. Numbered references or alphabetized references should be used instead.

Our reply

The references are listed at the end of the manuscript in the sequential order by which they are first mentioned in the main text. This is the format requested by the journal (<https://www.nature.com/nmeth/submission-guidelines/aip-and-formatting#references>).

Referee #3

One reference to the literature that should be included is O’Connell, Warner and Wang, 2001 Current Biology 1:702–707. This study uses a pair of micropipets (delivery-suction) to release cytochalasin D onto the surface of dividing NRK cells to examine localized actin dynamics at the cell cortex. The current work represents an advance over the system described 20 years ago, and the present manuscript provides both a detailed methodological description of the technique and sufficient control experiments to warrant publication in nature Methods. However, it would be helpful to contrast this current study with the technique used by the Wang group.

Our reply

We thank the referee for bringing this paper to our attention. The report from O'Connell *et al.* is a respectable and impressive effort, and one that we are happy to include in our discussion. Despite the similarity of their approach, their configuration does not allow the same performance as μ kiss.

In a nutshell, Wang *et al.* use a widely separated pipette-pair (20–40 μ m) across multiple dimensions. This renders an envelope of their injected material that is spatially extended. Such an envelope – significantly larger than that which we demonstrate – precludes microscale delivery areas, fast and prompt temporal response of establishing the envelope as well as diminishing the ability to confine reagents against diffusion-led escape.

Decision Letter, first revision:

Dear Vahid,

Thank you for submitting your revised manuscript "A paintbrush for delivery of nanoparticles and small molecules to live cells with micrometer spatial and millisecond temporal control" (N METH-A52872A).

We have now had a chance to go through the referee comments (below) and your proposed responses, and are writing to tell you we'll be happy in principle to publish it in Nature Methods, pending your described revisions to satisfy the referees' final requests and to comply with our editorial and formatting guidelines. When you resubmit, please include a point-by-point rebuttal describing the updates.

TRANSPARENT PEER REVIEW

Nature Methods offers a transparent peer review option for new original research manuscripts submitted from 17th February 2021. We encourage increased transparency in peer review by publishing the reviewer comments, author rebuttal letters and editorial decision letters if the authors agree. Such peer review material is made available as a supplementary peer review file. Please state in the cover letter 'I wish to participate in transparent peer review' if you want to opt in, or 'I do not wish to participate in transparent peer review' if you don't. Failure to state your preference will result in delays in accepting your manuscript for publication.

ORCID

IMPORTANT: Non-corresponding authors do not have to link their ORCIDs but are encouraged to do so. Please note that it will not be possible to add/modify ORCIDs at proof. Thus, please let your co-authors know that if they wish to have their ORCID added to the paper they must follow the procedure

described in the following link prior to acceptance:

Sincerely,

Rita

Rita Strack, Ph.D.

Senior Editor

Nature Methods

Reviewer #1 (Remarks to the Author):

After carefully read through the responses from the authors, I still found the following questions which were either not correctly answered or unanswered, leaving some fundamental issues of the proposed technique. Actually I did not see the made up experimental data with enough statically supports, which leave the numbers answered in vain. In more detail,

1. The current microkiss device mainly made of glass tubes pulled up to form the final channel with a tip size in a range of micron to ten microns, typically the breaking point will be very rough with a size variation larger than 30%, not the mentioned 10%, I think the authors need to provide the images of at least 100 real fabricated tips to calculate the variations.
2. The 2nd unanswered question is the fluctuation of the slender tip from the interference of the environment. The tip positions need to be recored for at least 10 minutes in both static or dynamic situations to see if the fluctuation is really smaller than 10% of the tip inner diameter.
3. To show the control ability of the microkiss device, for the same operation scheme, at least 10 times repeating experiments are needed to estimate the kissing area size and center positions are really within 10% variations.
4. Eventhouht the temporal resolution of microkiss can approach 35 ms mentioned in the response, however it is still much slower than the membrane protein response in ms to tens of microseconds, which can be easily achieved by micro/nsno fluidic systems.
5. Integrated micro/nano fluidic dose not necessary contain very tight wall with a dimension of cells, they can be freely and easily designed into a large chamber with a size more than mm, but nano channels connected to the bottom of the chamber close to cells can also be easily arranged, they can

also perform non-contact even more accurate/precise temporal/spatial control of the substance delivery ranging from molecule dyes to nano-particles. I feel the authors may not review enough literature regarding integrated micro/nano fluidic systems and state a very skewed response.

As a result, I feel the current manuscript is still far away from acceptable, and the authors seem would not like to provide more statically valid data to support their arguments, thus I recommend a rejection to the manuscript.

Reviewer #2 (Remarks to the Author):

The revised manuscript has addressed all the previously raised questions and concerns. As such, the recommendation is to publish the manuscript in this journal in its current form.

Reviewer #3 (Remarks to the Author):

The authors have thoughtfully addressed the comments presented by all the reviewers, both in the text, and through the inclusion of new experimental data. The strength of the method discussed here is that it is readily available to cell biologists/biophysicists/structural biologists possessing standard microscopy experiment and readily available micro-manipulators. Though several alternative methods are discussed, the authors point out that these may require more specialized equipment. It is for this reason that I would recommend publication. Micro-delivery/micro-injection can be dark arts, often restricted to the cognoscenti. As more fields of biology move towards single-cell experiments - sequencing, Cryo-EM tomography, etc., having readily available methods to manipulate individual cells, and having this accessible to diverse labs is well within the purview of Nature Methods and its readership.

Reviewer #4 (Remarks to the Author):

In the revised manuscript, the authors have addressed all the previously raised questions and concerns. As such, the recommendation is to publish the manuscript in this journal in its current form.

Author Rebuttal, first revision:

Referee Reply for NMETH-A5287A

We wish to extend our thanks to all the referees for their constructive and thought-provoking comments. The manuscript certainly stands much stronger as a result of their efforts. We are pleased that we have been able to address most of their concerns satisfactorily, and that they endorse our work for publication.

We address below several outstanding comments that require further clarification.

1. The current microkiss device mainly made of glass tubes pulled up to form the final channel with a tip size in a range of micron to ten microns, typically the breaking point will be very rough with a size variation larger than 30%, not the mentioned 10%, I think the authors need to provide the images of at least 100 real fabricated tips to calculate the variations.

We have not inspected 100 micropipettes, but we had indeed inspected 10 exemplar micropipettes via SEM imaging electron microscope, which resulted in our estimate of 10% variation. These images are explicitly included in Fig. 1 at the end of this letter. We can happily include all of these images alongside Figure 6 of the SI (Section 1.7) where we discuss the variability of the pipette tip sizes, provide numerical quantification of the variance and also simulate and discuss the (minimal) effect of the opening on the flow confinement.

The tips are provided commercially and as such they have a sufficiently tight tolerance of about 10%. We would take this opportunity to emphasize that the critical point is not that the fabrication process has specifically 10% tolerance, nor is it required to do so. Even if the tip opening varies by 30%, most experiments we perform or propose can be done successfully because the flow confinement does not depend as sensitively on the opening. Whether the manufacturer (in this case, Eppendorf) is so skilled to produce micropipette tips with such high precision at high yield is beside the point of our application. Finally, we should also like to point out that inspection of the micropipette tips for quality purposes in the laboratory is also not difficult because an opening of the order of 1-10 microns can also be measured with an optical microscope.

2. The 2nd unanswered question is the fluctuation of the slender tip from the interference of the environment. The tip positions need to be recorded for at least 10 minutes in both static or dynamic situations to see if the fluctuation is really smaller than 10% of the tip inner diameter.

In our first reply letter, we showed measurements that demonstrated the mechanical stability on the order of seconds because that is the relevant brushing time if one wants to deliver a small amount. Nevertheless, we have indirectly shown a long-term stability in Fig. 1h of the main manuscript. Here the linewidth of the written letters is below 10 micrometers. We nonetheless appreciate the concern raised, and have recorded the positional stability for the micropipette-pair over an extended period of 20 minutes, for the tips in μ kiss operation and with flow ceased. These results are shown below in Fig. 2, wherein one observes a sub-micron positional stability on the order of 100 nm.

3. To show the control ability of the microkiss device, for the same operation scheme, at least 10 times repeating experiments are needed to estimate the kissing area size and center positions are really within 10% variations.

Similar to the previous request, we are happy to provide further evidence of our μ kiss operation through repeated practice. We performed 10 back-to-back labeling events of CTxB-AF488 on the GM1-SLB system reported in Fig. 2 of the main manuscript. We show an illustrative still frame of one event in Fig. 3 below, and provide a video detailing the whole process (provided by attachment here electronically in a .gif format). We find, under operational parameters of $Q_{inj}=0.07 \text{ nLs}^{-1}$ and $Q_{exp}=2.54 \text{ nLs}^{-1}$ for micropipettes with $6 \mu\text{m}$ apertures, that we can label a region of the membrane with an average area of the $65.6 \pm 7.3 \mu\text{m}^2$ – corresponding to a standard deviation of 11.3%.

4. Eventhougt the temporal resolution of mikrokiss can approach 35 ms mentioned in the response, however it is still much slower than the membrane protein response in ms to tens of microseconds, which can be easily achieved by micro/nsno fluidic systems.

We remain not aware of any method that can perform labeling in the localized and 'near-field' manner we describe in the manuscript at the temporal response we achieve, namely ca. 35 ms. This assertion is also built on premise that it is not sufficient to merely demonstrate, for example, a fast micro/nano fluidic valve, if such a valve cannot be incorporated in a geometry and dimension to achieve microscale local exclusive delivery to the near-field space of individual cells. We would have greatly appreciated to receive concrete references demonstrating micro-nano fluidic systems that perform what we do at a higher speed, but are not aware that this has been achieved. Page 2 of 5

5. Integrated micro/nano fluidic dose not necessary contain very tight wall with a dimension of cells, they can be freely and easily designed into a large chamber with a size more than mm, but nano channels connected to the bottom of the chamber close to cells can also be easily arranged, they can also perform non-contact even more accurate/precise temporal/spatial control of the substance delivery ranging from molecule dyes to nano-particles. I feel the authors may not review enough literature regarding integrated micro/nano fluidic systems and state a very skewed response.

In continuation with our previous remark, it would have been perhaps more fruitful if the referee had guided us to specific literature they have in mind. We sympathize with their general feeling that it should be possible to fabricate interesting devices with micro/nano fluidic technology, but we politely disagree that such systems have been shown thus far to achieve the performance we demonstrate, and also that they would, in principle, encounter great technical challenges in order to match that which we have demonstrated. We also feel it is important to emphasize that it is most common in cell biology to work with an open cell culture, and as such our solution meets this immediate practical need.

Accompanying Figures

Figure 1 | 10 Scanning Electron Microscopy images of commercially-available micropipette tips with nominal aperture diameter of 6 µm.

Page 4 of 5

Figure 2 | Positional stability of micropipette-pair. **a**, histogram presenting the frame-to-frame variation in the position of the micropipette pair over the course of 20 mins, with a frame exposure time of 0.931 s for when flow is actuated. Inset shows a composite confocal iSCAT (gray scale) and fluorescence (green) image of the micropipette-pair with a fluorescently-labeled reagent under flow. Here, $Q_{inj}=0.07 \text{ nLs}^{-1}$ and $Q_{app}=2.96 \text{ nLs}^{-1}$. **b**, histogram showing the frame-to-frame variation in the position of the micropipette pair over the course of 20 mins, with a frame exposure time of 0.931 s for when flow is ceased.

Page 5 of 5

Figure 3 | Exemplary composite image of a μ kiss labeling event, viewed through confocal iSCAT microscopy (gray scale image) and fluorescence (green component overlay). A SLB membrane containing the GM1 lipid is μ kiss labelled by a 1 mg ml^{-1} fluorescent solution of CTxB-AF488, where $Q_{\text{inj}}=0.07\text{ nLs}^{-1}$ and $Q_{\text{asp}}=2.54\text{ nLs}^{-1}$ for micropipettes with $6\text{ }\mu\text{m}$ apertures. A full video of all 10 such back-to-back labeling events is provided. The area of the labelled region, across all 10 samples is on average $65.6 \pm 7.3\text{ }\mu\text{m}^2$.

Final Decision Letter:

Dear Vahid,

I am pleased to inform you that your Article, "A paintbrush for delivery of nanoparticles and small molecules to live cells with micrometer spatial and millisecond temporal control", has now been accepted for publication in Nature Methods. The received and accepted dates will be June 12, 2023 and Jan 8, 2024. This note is intended to let you know what to expect from us over the next month or so, and to let you know where to address any further questions.

Over the next few weeks, your paper will be copyedited to ensure that it conforms to Nature Methods style. Once your paper is typeset, you will receive an email with a link to choose the appropriate publishing options for your paper and our Author Services team will be in touch regarding any additional information that may be required. It is extremely important that you let us know now whether you will be difficult to contact over the next month. If this is the case, we ask that you send us the contact information (email, phone and fax) of someone who will be able to check the proofs and deal with any last-minute problems.

Please note that *Nature Methods* is a Transformative Journal (TJ). Authors may publish their research with us through the traditional subscription access route or make their paper immediately open access through payment of an article-processing charge (APC). Authors will not be required to make a final decision about access to their article until it has been accepted. [Find out more about Transformative Journals](https://www.springernature.com/gp/open-research/transformative-journals)

Authors may need to take specific actions to achieve [compliance with funder and institutional open access mandates](https://www.springernature.com/gp/open-research/funding/policy-compliance-faqs). If your research is supported by a funder that requires immediate open access (e.g. according to [Plan S principles](https://www.springernature.com/gp/open-research/plan-s-compliance)) then you should select the gold OA route, and we will direct you to the compliant route where possible. For authors selecting the subscription publication route, the journal's standard licensing terms will need to be accepted, including [self-archiving policies](https://www.springernature.com/gp/open-research/policies/journal-policies). Those licensing terms will supersede any other terms that the author or any third party may assert apply to any version of the manuscript.

If you are active on Twitter/X, please e-mail me your and your coauthors' handles so that we may tag you when the paper is published.

Best regards,
Rita

Rita Strack, Ph.D.
Senior Editor
Nature Methods